# BIRB: A GENERALIZATION BENCHMARK FOR INFORMATION RETRIEVAL IN BIOACOUSTICS

## ABSTRACT

The ability for a machine learning model to cope with differences in training and deployment conditions—e.g. in the presence of distribution shift or the generalization to new classes altogether—is crucial for real-world use cases. However, most empirical work in this area has focused on the image domain with artificial benchmarks constructed to measure individual aspects of generalization. We present BIRB: a generalization Benchmark for Information Retrieval in Bioacoustics,[1] a complex benchmark centered on the retrieval of bird vocalizations from passively-recorded datasets given focal recordings from a large citizen science corpus available for training. We propose a baseline system for this collection of tasks using representation learning and a nearest-centroid search. Our thorough empirical evaluation and analysis surfaces open research directions, suggesting that BIRB fills the need for a more realistic and complex benchmark to drive progress on robustness to distribution shifts and generalization of ML models.

## 1 INTRODUCTION

Generalization is one of the most important properties when deploying machine learning models: applications require models which perform reliably when shifting from curated training and evaluation data to real-world conditions. Generalization is difficult to study and includes many different factors (Moreno-Torres et al., 2012) including label shift (changes in the balance of labels between training and test time), covariate shift (which subsumes a vast array of data differences, including sensor or environmental changes), and few-shot learning (using a few examples to demonstrate generalization to novel concepts or classes). Previous works try to disentangle these aspects of model generalization by creating synthetic or artificial datasets. This leads to the development of approaches which handle only specific aspects of generalization, and which may not compose well into a workable solution for real-world data. Steady improvement on artificial datasets may thus give a false sense of overall progress. There is a need for datasets which allow us to study problems in model generalization holistically.

ML for bioacoustics has matured rapidly over recent years, and now includes a large and varied collection of freely available datasets which provide an ideal sandbox for studying model generalization. Birds are particularly well suited for this study as they include over $10,000$ species (most of which are highly vocal), are found in almost every habitat across the world in vastly different environments and ambient sound conditions, and are likely the world's most recorded group of animals.

Citizen science constitutes a large source of training data for bird vocalization detection, but training models on this data and deploying them for practical applications presents a rich collection of generalization challenges (Goëau et al., 2018; Kahl et al., 2021). Training recordings are often *focal* and are typically collected using directional microphones to acoustically focus on a particular bird, though other species may be present in the *background*. By contrast, models are deployed on *soundscapes*, i.e. *passive* recordings made with omnidirectional microphones capturing every sound occurring in the vicinity of the recorder, often with low signal-to-noise ratio. A range of passively recorded soundscape datasets from all over the world have recently been published, which are annotated by human experts and are typically small slices of unlabeled datasets which can comprise thousands or even millions of hours of audio (Roe et al., 2021; Lesmeister & Jenkins, 2022).

---

[1] https://www.audubon.org/news/when-bird-birb-extremely-important-guide

The difference in recording and deployment modalities between the crowdsourced citizen science and soundscape datasets present challenging *distribution shift* problems. As both dataset types reflect bird species in the wild, these naturally exhibit label imbalances depending on relative aspects of the populations present (population sizes, call frequencies and types, etc.). Furthermore, soundscape datasets are geographically localized and, consequently, only a small subset of species appear in each soundscape, producing drastic label shifts relative to globally-collected training data. Using these large-scale, high-quality, regional soundscapes, we take advantage of the opportunity to disentangle and measure a collection of generalization challenges across a diverse collection data and attempt to quantify the relative impact of these on model performance.

Conservationists use bioacoustics to address a wide range of high-impact environmental issues, including protecting highly endangered animals, detecting invasive species early, measuring broad biodiversity, and monitoring the impact of policies and interventions. Bioacoustics provides an incredibly rich picture of the natural world, but is challenging due to factors such as: overlapping vocalizations; intra-species variations; background noise, confounding biological and non-biological signals; and scarce training data for many of the world's most endangered species and fragile ecosystems. Despite the difficulty, bioacoustic deployments are rapidly growing in number and scale thanks to the ease of data collection.

As computational bioacoustics evolves, more sophisticated research questions that go beyond detecting the presence or absence of a species can be tackled with ML techniques. For example: "Which call types appear?", "What can these call types tell us about the behavioral state of the animals (e.g., feeding, breeding)?", "How many young does each adult have?", and so on. We can answer these questions by retrieving relevant vocalizations from the dataset, e.g., given a few examples of juvenile and adult vocalizations, one could retrieve relevant vocalizations from the dataset and calculate the ratio. Our benchmark is framed as a retrieval task, closely aligning with real-world use.

## 1.1 BENCHMARK OVERVIEW

We introduce BIRB: a generalization Benchmark for Information Retrieval in Bioacoustics [2], with both the bioacoustics and ML research communities to evaluate their models against a complex and real-world acoustic setting, study several facets of generalization in bioacoustics, and steer the community towards model development for more complex problems; (2) to support the needs of practitioners in diverse domains in processing large volumes of unlabeled audio field data.

BIRB is a **retrieval task**: first a model is trained using an upstream dataset, then during evaluation a handful of labeled vocalizations (exemplars) of a particular class are used to retrieve and rank all instances (vocalizations) of the same class in a given corpus. The benchmark measures the following challenges: **out-of-distribution generalization** (retrieving vocalizations from passive recordings after having trained on only focal recordings), **few-shot learning** ability (retrieving vocalizations of novel species given only a few instances of them), and robustness to **class imbalance and label shift** (both the upstream data and search corpora have a long-tailed class distribution, which can vary widely between the upstream data and search corpora).

Our baseline system (Section 3.2.1) allows any embedding model to be evaluated without needing the user to implement a few-shot learning algorithm. This is achieved using a nearest neighbor search over fixed embeddings, resulting in a scalable and efficient system which aligns with the constraints imposed by objective (2). This precludes a variety of interesting methods (e.g., adapting the model during evaluation using the exemplars) but this design decision supports our second objective of being able to handle very large amounts of audio data. Extending this baseline system to explore other approaches while maintaining computational efficiency is important future work.

## 2 RELATED WORK

As argued in Section 1, generalization in the presence of various differences between training and deployment conditions (label shift, covariate shift, novel classes, etc.) is arguably one of the most important open problems in deep learning, and that problem spans several areas of research.

---

[2]Our benchmark codebase is open-sourced under the Appache License 2.0 in GitHub.

Table 1: Summary of Bioacoustic Soundscape Dataset Characteristics. 'Eval(R)' indicates artificially rare species set, and 'Eval(H)' indicates a species set entirely held out from the training set.

| Dataset | Role | Hours | #Species | #Labels | Climate |
|---|---|---|---|---|---|
| Xeno-Canto (Vellinga & Planqué, 2015) | Train+Eval | >10k | >10k | >750k | Various |
| Pennsylvania, USA(Chronister et al., 2022) | Valid | 6 | 48 | 16,052 | Temperate |
| New York State, USA (Kahl et al., 2022a) | Eval(R) | 285 | 96 | 50,760 | Temperate |
| Island of Hawai'i, USA (Navine et al., 2022) | Eval(H) | 51 | 27 | 59,583 | Tropical |
| Colombia & Costa Rica (Vega-Hidalgo et al., 2023) | Eval(H) | 34 | 89 | 6,952 | Tropical |
| High Sierra Nevada, USA (Clapp et al., 2023) | Eval | 34 | 19 | 10,296 | Sub-Alpine |
| Sierra Nevada, USA (Kahl et al., 2022c) | Eval | 33 | 56 | 20,147 | Temperate |
| Peru (Hopping et al., 2022) | Eval | 21 | 132 | 14,798 | Tropical |

**Generalization:** Out-of-distribution (OOD) generalization goes beyond the traditional machine learning assumption that the training and target data are $i.i.d.$ Two areas which relax the $i.i.d.$ constraint to achieve model generalization in the presence of domain shift are *domain generalization* and *domain adaptation*; robust surveys of this field cover the standard methods and benchmarks (Wang & Deng, 2018; Wang et al., 2022; Zhou et al., 2022). Most of those benchmarks are quite removed from practical applications, but the WILDS benchmark (Koh et al., 2021) is directly motivated by the need to study real-world problems and robustness to naturally-occurring distribution shifts. Common domain generalization and domain adaptation evaluation protocols assume that the label space of the upstream training data and of the evaluation data match exactly.

The audio domain has received little attention so far in academic studies of generalization: the majority of prior benchmarks focus on vision and language modalities that don't extend to other important application areas. We note that simple direct application of methods developed for the vision domain may be insufficient for bioacoustics, based on empirical evidence from Boudiaf et al. (2023) which recently showed failure modes of translating these methods to audio tasks. In Section A.2 we provide further context relating BIRB to the wider literature on generalization.

**Audio & Bioacoustics:** Bioacoustics is a fast growing field (Bakker, 2022), with an expansive variety of problems suitable for machine learning (Stowell, 2022). Many of the datasets used in our benchmark have previously been released as part of BirdCLEF competitions (Goëau et al., 2018; Kahl et al., 2022b). This competition, similarly to our benchmark, provides participants with Xeno-Canto training data while evaluating models on soundscapes. We collect these datasets into a framework that allows for a more principled study of generalization.

Other recent bioacoustics competitions include a DCASE 2022 few-shot learning task and a Kaggle 2022 competition to identify birds and frogs in Puerto Rico organized by Rainforest Connection. BEANS is a recent benchmark that aims to measure the performance of ML techniques on a wide range of taxa (Hagiwara et al., 2022). Note that non-bird bioacoustics datasets tend to be a fraction of the size of avian datasets, both for training and evaluation.

The HEAR benchmark aims to surface audio embedding approaches which are generalizable to a wide variety of downstream tasks (Turian et al., 2022). Tasks in HEAR fall broadly into speech, music transcription, and audio event detection, and the initial benchmark competition found no single model which excelled in all three domains. Avian bioacoustics bears certain similarities with these domains, and while our our set of tasks is more narrowly scoped, we provide a more structured and rigorous platform for investigating factors in generalization performance.

# 3 MODEL GENERALIZATION BENCHMARK

## 3.1 DATASETS

This benchmark is built on a collection of publicly available datasets which we significantly preprocess and align into a cohesive mega-dataset.

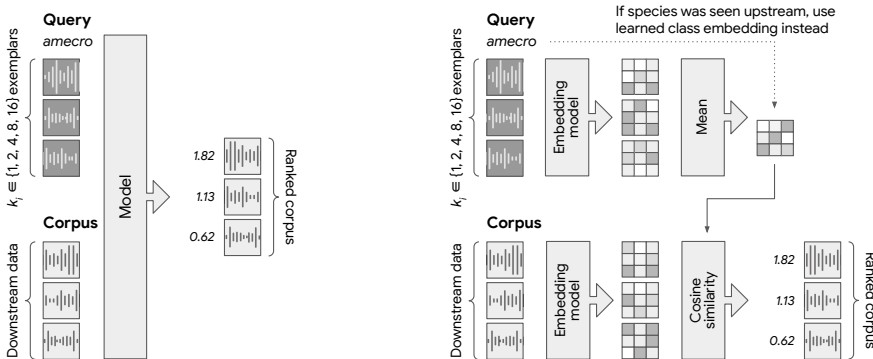

Figure 1: A schematic representation of the evaluation procedure (left) and our baseline system (right). Note that *amecro* is the species code for the American crow and is part of the query.

Xeno-Canto (XC; Vellinga & Planqué, 2015), a citizen science project, is the world's largest openly accessible repository of recordings of bird vocalizations, with more than $750,000$ recordings of over $10,000$ species (Table 1, first row). XC is comparable in scale to AudioSet-2M (the largest commonly used audio event detection benchmark; Gemmeke et al., 2017). Each recording is weakly-annotated with a single *focal species* tag; thanks to community discussion and rating systems, the focal species label is generally, but likely not always, correct. As Xeno-Canto is a crowdsourced dataset, it contains many imbalances. Recordings vary in length between seconds and hours, and contain significant gaps between vocalizations of the focal species. An optional list of *background species* may be provided, which may be incomplete or entirely absent for any particular recording.

We leverage a collection of passively recorded soundscape datasets obtained from specific ecological monitoring programs (Table 1, subsequent rows). Each of these datasets contains fine-grained time-frequency box species labels of all bird vocalizations. Because birds tend to sing at similar times of the day (e.g. dawn chorus), time-frequency boxes often overlap. Furthermore, the label distribution is imbalanced and not necessarily correlated with prevalence of XC examples. All soundscape datasets except the one recorded in Pennsylvania, USA were released following inclusion in BirdCLEF audio classification challenges, such as (Kahl et al., 2022b). Each of these soundscape datasets represents hundreds of hours of expert annotation effort.

Adapting and aligning this collection of datasets involves: (1) resolving significant differences in species taxonomies used across datasets; (2) correctly and consistently aligning labels in various input formats; (3) extracting fixed-length slices from file-level labels/annotations using peak-finding; and (4) converting timeboxed annotations in soundscape data into fixed-length labeled slices.

## 3.2 TASK

We provide upstream data from Xeno-Canto with which practitioners can produce a model. Since the original Xeno-Canto recordings are of variable length, we extract up to $5$ windows of 6 seconds each from each recording using a peak-finding algorithm (see Appendix A.3.1).

Models are then evaluated on downstream retrieval tasks which are represented by (1) a query, i.e. a species code along with a handful of example vocalizations (**exemplars**), and (2) a corpus from which to retrieve vocalizations. The goal is to retrieve vocalizations of the query species. For each retrieval task, models rank candidates in the corpus according to their relevance to the exemplars (Figure 1, left). We repeat the retrieval task over multiple random exemplar sets of size $k_i \in \{1, 2, 4, 8, 16\}$. The quality of the retrieved results is measured by the ROC-AUC metric, as discussed in Section 4.2.

### 3.2.1 BASELINE SYSTEM

There are many design choices left open to a user of our generalization benchmark. For the purposes of the empirical evaluation we present in Section 4 and given the constraints outlined in Section 1.1, we utilize the following protocol to carry out retrieval for each of our baseline models:

First, acquire an **embedding model** via training on the large, curated **upstream** dataset or a pre-trained one with a transferable learned representation; this model is used to embed the **evaluation** datasets of query species and candidate corpora. Our general framework's primary constraint is the upstream data availability; we allow the architecture, learning algorithm, and other considerations to be design choices made by the user.

Second, we select a method for transforming embeddings into one or more queries for each species: we do this by averaging the embeddings for a given species, but many approaches could be explored. To carry out the comparison between query and candidate, we use the cosine similarity metric, though other metrics (including distances) are supported. The cosine nearest-centroid approach was previously shown to be a strong few-shot learning baseline (Chen et al., 2021).

Finally, we iterate over the various queries and candidate corpora and rank the scores, over which we compute metrics to measure the quality of the retrieved instances.

### 3.3 DATA CONSTRUCTIONS

To incorporate and isolate some core real-world complexities in this setting, we provide the following constructions to produce the **upstream** data and **evaluation** data. Classes included in the upstream data are those which we allow to be available at the time of producing an embedding model (such as during training).

The evaluation data is disjoint from the upstream set and consists of multiple datasets and classes: (1) heldout examples from classes and data present in the upstream dataset (artificially rare classes); (2) examples from classes in the upstream data in different recording settings (covariate shift); (3) classes withheld from the upstream data (novel classes); (4) additional soundscape datasets from different geographical regions (label shift).

#### 3.3.1 UPSTREAM DATA

Our upstream data is the entire XC dataset (excluding restricted species) with the following caveats.

**Artificially Rare Data:** As our data is composed of naturally occurring animal species which can experience fluctuations in population (particularly decreases), and for which "rarity" is loosely correlated with bird populations. We present an "artificially rare" set derived from the dataset recorded in New York State, USA. This task captures the challenge of *robustness to class imbalance*. For each species in the dataset, we randomly sample 10 recordings (from which up to 50 windows are extracted using peak finding) in which that species appears in the focal annotation to include in the upstream data, reserving the species' other focal recordings from this region for evaluation.

**Heldout Eval Regions:** This dataset construction corresponds to our **novel-class retrieval** task. From the upstream data, we *exclude* all recordings of species present in the Colombia & Costa Rica and Island of Hawai'i, USA datasets to capture model generalization to novel or unobserved species. Note that while we exclude recordings with explicit focal or background recordings, it is possible that a species vocalization may be present in some instances in an unlabeled manner. We include all recordings for these species as part of our XC Focal corpora in the evaluation data.

#### 3.3.2 EVALUATION DATA

The evaluation data forms the basis of the retrieval task and is composed of the data reserved from XC data from the artificially rare (New York State, USA) set and heldout regions (Island of Hawai'i, USA and Colombia & Costa Rica), along with soundscape datasets for each region (New York State, USA, Island of Hawai'i, USA, Colombia & Costa Rica, High Sierras, USA, Sierra Nevada, USA, and Peru). In our large-scale evaluation, we embed all of the evaluation data using each of our baseline embedding models.

**Queries:** We use peak finding (Appendix A.3.1) to draw slices from XC recordings with focal labels for each species of interest to form the basis of the query. These are used to compare with candidates from focal and soundscape corpora.

**Candidate Corpora:** For each regional evaluation set, candidates are drawn from XC using recordings with focal labels. Additionally, we compose Soundscape corpora for each region using the datasets described in Table 1.

## 4    LARGE-SCALE EMPIRICAL EVALUATION

We evaluate our baseline system (Section 3.2.1) with a variety of embedding models, including linear and deep learning architectures, and models pretrained on AudioSet. Amongst the deep learning models, we compare a mix of convolutional and transformer-based architectures, and also compare models trained specifically on birdsong against off-the-shelf general audio event detection models. Our models are trained using standard empirical risk minimization and hyperparameter tuning, which is a strong baseline for domain generalization (Gulrajani & Lopez-Paz, 2020).

### 4.1    BASELINE EMBEDDING MODELS

Our linear embedding models are trained on handcrafted features extracted from mel spectrograms. We experimented with several features, including the MFCC statistics used by the BEANS benchmark (Hagiwara et al., 2022), find that applying an average-pooling operation to the Mel spectrogram representation and flattening the output provided reasonable results (denoted by "Pooled Melspecs").

We consider four deep embedding models trained for classification of bird species, two sizes of EfficientNet v1 (B1 and B5, denoted by "EfficientNet S" and "EfficientNet L" resp.) (Tan & Le, 2019) and two sizes of Conformer architectures (Gulati et al., 2020) (denoted by "Conformer S" and "Conformer L" resp.). These models are trained similarly, differing only in the model architecture, following the recipe in Denton et al. (2022), and described in Appendix A.3.3. Note that we perform hyperparameter sweeps on a validation soundscape (Pennsylvania, USA) and we use the activations preceding the classification layer as a general embedding of birdsong.

Finally, we consider embeddings from pre-trained models trained on AudioSet (Gemmeke et al., 2017). YAMNet[3] is a general audio event detection classifier using a convolutional architecture. AudioMAE (Huang et al., 2022) is a self-supervised transformer, trained to reconstruct masked spectrograms. We evaluated a re-implementation of the 'Large' AudioMAE model with 300M parameters. This model obtains a mean average precision (mAP) of 46.4 on AudioSet-2M after fine-tuning, comparable to the original AudioMAE's reported mAP of 47.3.

### 4.2    METRICS

Retrieval is a significant use-case for bioacoustics models, enabling us to understand the presence and prevalence of different sounds, like vocalizations, in large unlabeled audio data. We align with previous bioacoustics works (Stowell, 2022; Sebastián-González et al., 2015) and score the quality of the ranked list of results returned by the model using the area under the receiver operating characteristic curve (ROC-AUC), a threshold-free metric similar to average precision. Given a model $f$ and sets of positives $D^+$ and negatives $D^-$ the ROC-AUC is calculated as

$$\text{ROC-AUC}(f) = \frac{\sum_{x^+ \in D^+} \sum_{x^- \in D^-} [f(x^+) < f(x^-)]}{|D^+||D^-|}.$$

We choose ROC-AUC over average precision because it produces scores that can be compared across datasets and queries (species) which have differing numbers of positives and negatives.[4] Although ROC-AUC is often presented as a metric to evaluate binary classifiers, it is in fact equivalent to the probability that a random pair of positive and negative examples is ranked correctly.

We denote by **cROC-AUC** the ROC-AUC scores averaged across classes (species). We use the geometric mean for averaging to emphasize that we care about the relative improvement across

---

[3]https://github.com/tensorflow/models/tree/master/research/audioset/yamnet

[4]For example, a random ranking of 1 positive and $n$ negatives has an expected average precision of $\frac{H_{n+1}}{n+1}$, whereas the expected value of ROC-AUC for a random ranking is always 0.5.

Table 2: Geometric cROC-AUC results for the Artificially Rare and Heldout datasets. ROC-AUC scores for each species are averaged over 5 sampled sets of exemplars.

| Corpus | Model | New York State, USA | | | Colombia & Costa Rica | | | Island of Hawai'i, USA | | |
|---|---|---|---|---|---|---|---|---|---|---|
| | | $k = 1$ | 8 | 16 | 1 | 8 | 16 | 1 | 8 | 16 |
| XC Focal | Pooled Melspecs | 0.57 | 0.61 | 0.62 | 0.58 | 0.63 | 0.64 | 0.58 | 0.62 | 0.60 |
| | EfficientNet S | 0.75 | 0.83 | 0.84 | 0.78 | 0.89 | 0.90 | 0.79 | 0.90 | 0.91 |
| | EfficientNet L | 0.75 | 0.81 | 0.81 | 0.77 | 0.85 | 0.86 | 0.77 | 0.87 | 0.87 |
| | Conformer S | 0.76 | **0.86** | **0.87** | **0.79** | **0.89** | **0.90** | **0.82** | **0.93** | **0.93** |
| | Conformer L | **0.76** | 0.83 | 0.84 | 0.77 | 0.88 | 0.89 | 0.81 | 0.90 | 0.91 |
| | AudioMAE | 0.52 | 0.52 | 0.53 | 0.53 | 0.55 | 0.56 | 0.56 | 0.59 | 0.58 |
| | YamNet | 0.57 | 0.59 | 0.59 | 0.57 | 0.61 | 0.62 | 0.62 | 0.66 | 0.65 |
| Soundscapes | Pooled Melspecs | 0.50 | 0.51 | 0.51 | 0.51 | 0.53 | 0.53 | 0.45 | 0.46 | 0.47 |
| | EfficientNet S | 0.63 | 0.72 | 0.73 | 0.67 | **0.79** | **0.81** | 0.64 | 0.73 | 0.73 |
| | EfficientNet L | 0.62 | 0.70 | 0.71 | **0.67** | 0.78 | 0.79 | 0.65 | 0.73 | 0.73 |
| | Conformer S | **0.65** | **0.74** | **0.75** | 0.67 | 0.76 | 0.79 | **0.66** | **0.76** | **0.77** |
| | Conformer L | 0.64 | 0.70 | 0.71 | 0.63 | 0.73 | 0.75 | 0.63 | 0.72 | 0.73 |
| | AudioMAE | 0.49 | 0.49 | 0.49 | 0.47 | 0.46 | 0.45 | 0.46 | 0.47 | 0.49 |
| | YamNet | 0.49 | 0.50 | 0.50 | 0.51 | 0.51 | 0.51 | 0.43 | 0.45 | 0.50 |

species (i.e. increasing the ROC-AUC of a query from 0.5 to 0.6 is more important than increasing a score from 0.8 to 0.9; Voorhees, 2005, Section 4.2). Using geometric averaging also allows us to calculate relative improvements across models (Fuhr, 2018, Section 2.4). Models are evaluated using a variable number of $k$ exemplars, as reported in Tables 2 and 3.

### 4.3 GENERALIZATION RESULTS

We evaluate the upstream-trained models and pre-trained baselines described in Section 4.1 on BIRB and present the results per corpus in Tables 2 and 3. In the Appendix we additionally provide figures with all examplar set sizes, along with results for each species.

To contextualize our baseline results, we highlight the generalization challenges each task reflects, noting that each contains a subtask on covariate shift in comparing retrieval performance on XC Focal vs. the Soundscape counterpart. We recall our method for producing queries: we embed the $k$ exemplars, compute their mean, and use this as the query for that species. We intentionally opt for a simple method to demonstrate the room for exploring more thoughtful choices which could likely improve retrieval quality. Similarly, to investigate the value of including Artificially Rare (AR) species in upstream model training, we extract the learned representations and average these in the same manner to produce each species' query.

**Artificially Rare Species:** This task aims to capture model robustness to class imbalance and classes with low representation. We compare performance when using exemplars vs. the learned representation of the Artificially Rare species in Figure 2; from these results, each model benefits from taking advantage of the learned representation when the species was available during embedding model training, providing improved performance over using the exemplars as the query. Using the learned representation is not a viable approach in many applications (i.e. a learned representation is not available for many of our evaluation regions or species). The gap suggests that further exploration of techniques for using exemplars to score retrieval candidates could improve model performance.

In comparing AR vs. the Heldout Regions, the AR species don't appear to benefit from the inclusion of a small number of training examples upstream. Disentangling why presents an area for further investigation: this could be attributed to class imbalance, but this behavior could also be indicative of the intrinsic difficulties of the New York State, USA dataset itself.

**Heldout Regions:** This task addresses the question of whether models can generalize to novel classes, i.e. retrieve species that were unavailable in the upstream data, aligned with the novel-class retrieval task described in Section 3.3. From our evaluation on Island of Hawai'i, USA and Colombia & Costa Rica, we observe that the deep models trained on our upstream XC data (EfficientNet and Conformer models) show appreciable retrieval capabilities for new classes. Our empirical investigation, however, does not demonstrate that the larger variants of the EfficientNet and Conformer

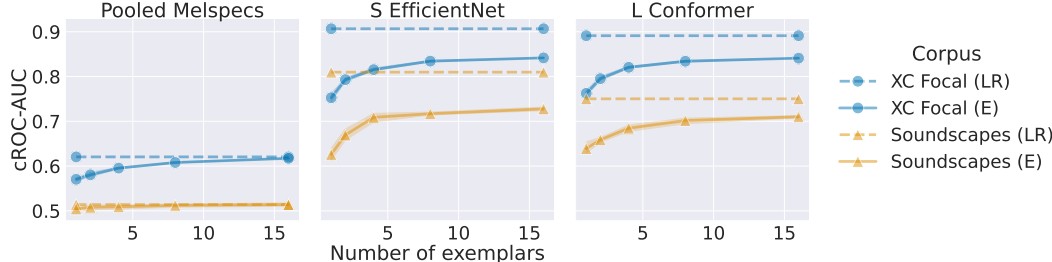

Figure 2: A comparison of using exemplars (E), i.e. $k \in \{1, 2, ...\}$, and the learned representation (LR) for Artificially Rare species from the New York State, USA region. The learned representation is derived from each model's learned weight vector for that species.

architectures provide significant benefit over their smaller counterparts, an interesting area for further investigation.

**Comparison between XC & AudioSet pre-training:** AudioMAE and YAMNet are both pre-trained on the AudioSet dataset (self-supervised and supervised, resp.). Out of the box, we observe a significant difference in performance when compared to our XC upstream-trained deep models, and the former often perform comparably with Pooled Melspecs, which does not incorporate any learning. We are careful to note that this is not reflective of the AudioMAE and YAMNet models' best possible performance, but rather that this indicates pre-training on general acoustic data is insufficient in the absence of some form of adaptation using problem-specific (i.e. XC upstream) data. Doing so (via transfer learning, domain adaptation, etc.) constitutes an interesting open direction to explore.

**Robustness to distribution shift:** In agreement with prior works (Goëau et al., 2018; Kahl et al., 2021), we observe a significant and consistent performance gap between XC Focal and Soundscapes corpora, indicating that all baselines suffer from poor robustness to distribution shift.

**Performance in mixed conditions:** Table 3 shows evaluation results for the Sierra Nevada, USA, High Sierras, USA, and Peru Soundscape datasets that consist of a mixture of species available (abundantly or in the Artificially Rare set) and heldout from the upstream data. In terms of cROC-AUC, models generally perform comparably or slightly worse compared to the other Soundscape datasets we evaluate in Table 2. When investigating performance on these regions partitioned into species sets by upstream availability, we observe a higher degree of variance when the species are heldout. Refer to Appendix A.4 for further discussion.

**Disentangling Label and Covariate Shift:** We construct an experiment where the full Xeno-Canto dataset is split into (1) a training dataset, (2) an $i.i.d.$ evaluation dataset which shares the training dataset's class distribution, and (3) a *label-shifted* evaluation dataset which shares Pennsylvania, USA's class distribution. We use Pennsylvania, USA for these ablations to align with the soundscape dataset used for validation in the main experiments. We also sample a small $i.i.d.$ (same class distribution) subset of the training dataset to compute training metrics on-the-fly.

In Figure 3 (left) we present cROC-AUC curves for evaluation, label-shifted evaluation, validation, and training $i.i.d.$ subset for the Conformer S architecture. We observe that the relative effect of

Table 3: Geometric cROC-AUC for additional soundscape eval sets, averaged as in Table 2.

| Model | Sierra Nevada, USA $k=1$ | 8 | 16 | High Sierras, USA 1 | 8 | 16 | Peru 1 | 8 | 16 |
|---|---|---|---|---|---|---|---|---|---|
| Pooled Melspecs | 0.50 | 0.53 | 0.52 | 0.48 | 0.48 | 0.48 | 0.47 | 0.46 | 0.46 |
| EfficientNet S | 0.58 | 0.63 | 0.63 | **0.66** | **0.77** | **0.79** | 0.57 | 0.62 | 0.62 |
| EfficientNet L | 0.57 | 0.59 | 0.60 | 0.64 | 0.73 | 0.73 | **0.59** | **0.64** | **0.65** |
| Conformer S | **0.59** | **0.66** | **0.66** | 0.65 | 0.75 | 0.76 | 0.56 | 0.61 | 0.61 |
| Conformer L | 0.59 | 0.62 | 0.63 | 0.64 | 0.68 | 0.70 | 0.53 | 0.56 | 0.56 |
| AudioMAE | 0.44 | 0.43 | 0.43 | 0.44 | 0.43 | 0.43 | 0.47 | 0.48 | 0.48 |
| YamNet | 0.40 | 0.41 | 0.41 | 0.39 | 0.43 | 0.45 | 0.49 | 0.49 | 0.49 |

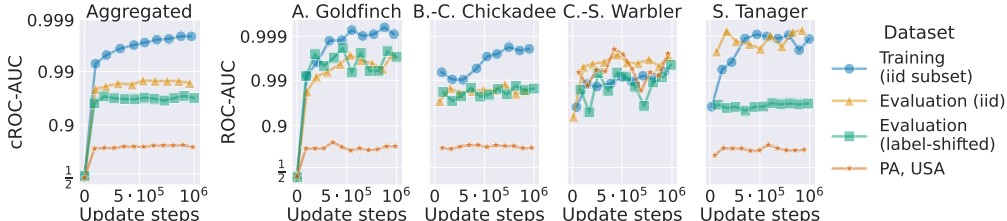

Figure 3: cROC-AUCs and ROC-AUCs on select species (in logit scale) throughout training on the Pennsylvania, USA dataset and various subsets of the XC dataset.

covariate shift is greater than that of label shift. However, ROC-AUC curves for individual species (Figure 3, right) show that the generalization challenge presented by the Xeno-Canto to soundscapes transfer problem is multi-faceted, as we at times observe no generalization gap (Chestnut-sided Warbler) and at other times observe generalization gaps that can be attributed to varied causes like covariate shift (American Goldfinch), i.i.d. generalization (Black-capped Chickadee), and label shift (Scarlet Tanager). Refer to Appendix A.4.1 for results on additional network architectures.

## 5 LIMITATIONS & FUTURE DIRECTIONS

Our thorough investigation surfaces a number of open challenges for researchers in retrieval, domain adaptation and transfer learning. Our large-scale evaluation reflects the performance of XC pre-trained embeddings and is not indicative of the performance one could achieve by adapting these on the exemplars; this highlights substantial possibilities worth exploring, e.g. within the context of domain adaptation, transfer learning, etc. Our evaluation datasets are annotated with timeboxes which are turned into overlapping fixed-size windows. This is consistent with established evaluation practices, but doing so excludes some families of approaches which could leverage longer and/or variable-length context. Future work enabling direct comparison with those approaches could unlock further improvements. Additionally, our baseline system does not currently incorporate metadata such as elevation, time of day or year, geographical location, etc., but BIRB's flexibility allows these to be integrated, which could offer improvements in model performance.

## 6 CONCLUSION

Model generalization is an open and foundational problem space; advancing this research benefits not only the ML community but translates to many bioacoustics and biodiversity problems. BIRB's methodology separates producing an embedding model using the upstream data with the constrained retrieval problem and evaluation. This allows pre-trained models to be used with or without adaptation, providing flexibility to ML researchers while enabling ease of use and adoption for domain experts. Based on our investigation, we find that the shift from focal to passive recordings contributes substantial generalizability challenges for embedding models, surprisingly even when increasing model size by a significant factor. Beyond covariate shift, our empirical results indicate that increasing the size of architectures we evaluate does not provide improved generalization in terms of label shift, robustness to distribution shift, and performance on novel species. While further study is needed, our intuition is that despite being fairly large, the XC dataset we train on is highly class-imbalanced and a significant portion of the bird species (classes) reflect a medium-to-low data regime that is not conducive to larger network architectures.

Our work is enabled by the open availability of comprehensive bioacoustic datasets; we redistribute these and provide our open-sourced codebase[5] to support adoption of BIRB and the development of ML research in this setting. We expect that insights generated by BIRB will translate directly to the improvement of ongoing acoustic monitoring efforts and directly aid UN Sustainable Development Goals for biodiversity and the environment. At the same time, we expect the complex, multifaceted generalization challenges in our benchmark to also innovate research in representation learning, domain adaptation, few-shot learning, and other areas.

---

[5]We will provide a link to our GitHub codebase after the review process ends.

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

## A APPENDIX

### A.1 WHY AVIAN BIOACOUSTICS?

**Challenges** Avian vocalizations, particularly bird songs, exhibit remarkable complexity, characterized by a rapid succession of elements. This complexity arises from the sophisticated vocal tract possessed by birds, enabling them to produce two-voiced sounds. The evolution of bird song involves a multifaceted process, often involving extensive learning, imitation, and improvisation, particularly in passerine birds. Within a species, the song repertoires can display immense diversity, ranging from a few to hundreds of distinct songs per individual. As an example, the brown thrasher, *Toxostoma rufum*, is known to produce over $1,000$ different song types, including imitations of other bird species. Additionally, the presence of local dialects further amplifies the complexity of species-specific vocalizations, posing challenges for bird identification based on sound and necessitating considerable learning efforts. Furthermore, birds frequently engage in simultaneous vocalization, particularly during the dawn chorus, a natural phenomenon occurring in the early morning hours. Consequently, the identification of bird sounds presents a multi-class, multi-label problem space. We recommend Pieplow (2019) as a general introduction to the intricacies of bird vocalizations.

**Data & Modeling Opportunities** The abundance of birds globally and across habitats, along with most birds being highly vocal, makes them particularly suitable for bioacoustics and ML research, as recording vocalizations is a much simpler task than collecting other data modalities (e.g. images in the wild, sensors, satellites). In addition, birds are found almost everywhere in the world and in almost every habitat, featuring vastly different ambient sound conditions (background noise levels, confounding biological and non-biological signals, etc.). These challenges offer rich opportunities for developing robust ML algorithms capable of handling complex and noisy acoustic environments. Lastly, birds are very accessible and have been recorded for many decades, making them likely the most recorded group of animals on the planet. Researchers can leverage this wealth of historical data to build robust and generalizable models that can recognize and classify various bird vocalizations accurately. Exploring and modeling bird vocalizations offer promising avenues for advancing our understanding of avian communication, species identification, and environmental monitoring using cutting-edge ML techniques.

Avian research frequently relies on digital tools for sound transformation. Spectrograms, which have a long-lasting tradition in bird biology, serve as visual representations of bird vocalizations and are extensively utilized for the analysis of audio recordings. It can be inferred that spectrograms hold valuable information regarding species identification, making them an ideal representation of avian sounds. Consequently, the most successful bird classification systems convert audio into a mel-spectrogram, and then apply networks originally developed for computer vision. An excellent survey of computational bioacoustics can be found in Stowell (2022).

### A.1.1 Acknowledgement of Potential Misuse

Given an extremely motivated actor, one could use the BIRB benchmark to identify vulnerable species for harassment or poaching. The process is elaborate, and we withhold details for how to engage in such behavior so as not to encourage misuse of this system which otherwise supports academic research and practical biodiversity and conservation work.

## A.2 Related Work

As introduced in Section 2, there are many fields focused on developing an understanding and effective methods for addressing different dimensions of the broader generalization problem.

A variety of areas and approaches exist to investigate generalization phenomena and improve model performance in their presence, such as transfer learning (Pan & Yang, 2010), meta-learning (Thrun & Pratt, 1998; Hospedales et al., 2021), self-supervised learning (first introduced by Bromley et al. (1993) and formalized by Chopra et al. (2005)), and semi-supervised learning (Chapelle et al., 2009). As explained in Section 3.2.1, the baselines investigated in our work all adopt the simple transfer learning principle of reusing a model trained on upstream data without further adaptation to embed evaluation data, as exemplified in Chen et al. (2019).

*Transfer learning* is a widely explored learning principle that uses learned knowledge from a previous task on a subsequent task. In the context of few-shot classification, a simple recipe is to use supervised training on the upstream dataset to produce a learned representation, then replace and learn a new classification layer on the subsequent task using the learned features (Chen et al., 2019) (with or without additionally fine-tuning the learned feature representation during that process (Kolesnikov et al., 2020)). Recent work has also proposed the use of intermediate features for transfer (Evci et al., 2022).

*Few-shot learning* studies the ability of a model to generalize to *new* classes from only a few labeled examples; Song et al. (2023) surveys few-shot learning approaches and standard benchmarks. Few-shot learning benchmarks were originally built by dividing the classes of a given dataset into ones that can be used for training and ones that are held-out for evaluation (Lake et al., 2013; Vinyals et al., 2016). This setup leads to minor (if any) covariate shifts between the upstream and evaluation datasets. The introduction of cross-domain few-shot learning benchmarks mitigates this drawback by studying generalization to novel classes that may originate from previously-unobserved datasets or distributions (Guo et al., 2020; Chen et al., 2019; Triantafillou et al., 2020; Dumoulin et al., 2021). While more complex, these efforts exclude other aspects of difficulty like class imbalance, which is prevalent in real-world problems. They also assume "novel classes" are not at all present in the

training data, whereas in real-world applications new classes may be present in the upstream data, but either very infrequently compared to other classes and/or in an unlabelled fashion.

*Meta-learning*, sometimes called "learning to learn", is a general strategy that can be applied to domain generalization, domain adaptation, or few-shot learning by using information from previous learning tasks to learn the learning algorithm itself (Thrun & Pratt, 1998). Metric learning-based meta-learning approaches (Vinyals et al., 2016; Snell et al., 2017) rely on comparisons in embedding space to make classification predictions, while optimization-based meta-learning methods instead focus on learning an initialization (Finn et al., 2017), or additionally a training algorithm (Ravi & Larochelle, 2017) for adapting a model to a new task (e.g. learning new classes from few examples). We refer the reader to (Hospedales et al., 2021) for a survey.

*Self-supervised learning* represents another avenue to learning a generalizable representation, via utilizing massive amounts of unlabeled data. Within computer vision, contrastive learning has demonstrated significant data efficiency and generalization, and there is work that theoretically investigates the generalization ability of this methodology (Huang et al., 2023). Moreover, recent empirical results have shown the power of self-supervision in learning rich representations for downstream tasks (Ericsson et al., 2022) and assist in labeling large unsupervised datasets.

*Semi-supervised learning* offers a way to harness both labeled and unlabeled data to further the predictive power of supervised models using transductive and inductive learning. Mix-Match (Berthelot et al., 2019) combines dominant approaches in this domain with the label augmentation method MixUp to obtain state-of-the-art results with their approach on image benchmarks like CIFAR-10 and STL-10.

Generally, artificially-constructed generalization benchmarks suffer from a fundamental limitation: it is unclear whether the type and degree of difficulty that they introduce (e.g. controlled by the choice of which classes or datasets to use upstream or for evaluation, how many labelled examples to provide for target classes, etc.) reflects realistic scenarios, thus potentially failing to steer research advances towards *relevant* problems. BIRB addresses this by introducing complex research challenges directly inspired by an important real-world problem.

## A.3 ADDITIONAL BENCHMARK TASK DETAILS FOR REPRODUCIBILITY

Table 4 contains a high-level summary of BIRB baseline model training details.

### A.3.1 DATASETS & PREPREPROCESSING

Across all XC datasets used within our framework, we exclude "restricted" recordings from the corpora. Restricted recordings reflect species which are at risk of poaching or harassment and are unavailable to the public (refer to XC's FAQ). Note that XC data is released under the Creative Commons license (CC); we redistribute this data under the CC-BY license. The soundscape data used in this work is hosted on Zenodo under CC licenses, described in our data and benchmark README (which we also redistribute).

Island of Hawai'i, USA and Colombia & Costa Rica represent new geographical regions composed of a collection of endemic species within each region. These species' class labels have not been presented to the model at training time, though there is a possibility of vocalizations *without labels* appearing in training data recordings.

Labels & Species Codes

Animal taxonomies are neither static nor unique, which complexifies machine learning on ecological data. For instance, XC uses the IOC World Bird List (updated every six months) as its taxonomy. Meanwhile, eBird and the Macaulay Library—and consequently the Cornell Lab of Ornithology, which is the main provider of our evaluation data—use the eBird taxonomy based on the Cornell Clements Checklist (updated in August each year). Both taxonomies have high overlap but differ for some key species.

Taxonomy updates contain changes like renaming species, splitting or merging species, and adding new species. Species names or codes therefore cannot be considered unique categorical identifiers. From a machine learning perspective, this has little impact so long as the species codes are matched

between the upstream and evaluation data and are kept consistent over time. From an application perspective, however, taxonomy changes pose a challenge and the real-world applicability of trained models may be limited if they rely on an outdated taxonomy.

The BIRB codebase standardizes around the 2021 eBird taxonomy and contains annotations at the species level, excluding sub-species labels. At the time of acquiring XC data (July, 2022), the website was using the IOC 10.1 taxonomy (it has since moved to the IOC 11.2 taxonomy). We mapped the IOC 10.1 species present on XC to eBird 2021 species code using a semi-automated process. The resulting mapping is stored in a serialized pandas DataFrame,[6] along with a list of XC recordings for each 2021 eBird species code. The mapping is also reflected in how hosted recordings are grouped into subdirectories by eBird 2021 species.[7] The mapping however is not perfect, and in particular there are four species which we decided to ignore altogether: *Vireo olivaceus* (reevir1), *Anas crecca* (gnwtea), *Cyanocorax yncas* (grnjay), and *Xiphorhynchus guttatus* (butwoo1).

Peak-finding

The Xeno-Canto dataset consists of recordings with lengths ranging from several seconds to over an hour. To simplify the training of models we pre-processed the dataset and extracted fixed-length slices of 6 seconds in length from each recording using a peak finding algorithm. To avoid significantly changing the label distribution of the original dataset we extract a maximum of 5 slices per recording. Peak finding proceeds as follows:

A mel-spectrogram of the recording is constructed using a window size of 80 ms and a hop of 10ms. The magnitudes are log-scaled (using a floor of $0.01$) and then scaled by $0.1$.

Then, a two-step denoising process is applied: For each frequency bin the mean and standard deviation of the log-magnitudes is calculated across time. Any values which are greater than the mean plus 1.5 standard deviations are discarded. A second mean and standard deviation is calculated using the remaining values. This second mean and standard deviation are used to select signal, which are all values that lie above the mean plus 0.75 standard deviations. The signal is shifted by this second mean, and the rest is discarded.

Finally, the magnitudes of the denoised spectrogram are summed across the frequency bins. SciPy's `signal.find_peaks_cwt` is then used to find peaks ranging between $0.5$ and $2$ seconds using 10 wavelet filters. We calculate the total value of the summed magnitudes in the 600 ms window surrounding the peak. If this total value is less than $1.5$ times the mean frequency-summed magnitude over the entire recording the peak is discarded. The remaining peaks are sorted by their summed magnitudes and only the top 5 are kept.

If a recording is less than 6s, it is padded with zeros before applying the peak finding. If not a single peak is found, we simply select the first 6s of the recording.

The full implementation can be found in the `find_peaks_from_audio` function in the `birb.audio_utils` module of the accompanying code.

Species exemplars for queries

The species exemplars used to form the queries in evaluation retrieval tasks are drawn from the peak-finding-processed XC dataset. Depending on the species' status (artificially-rare, heldout, fully available), exemplars are drawn from either the upstream or the downstream split:

- **Artificially rare** species exemplars are drawn from the ten recordings made available in the upstream split.
- **Heldout** species exemplars are drawn from the evaluation XC split; recall that no recordings are made available in the upstream XC dataset.
- **Fully available** species exemplars are drawn from the upstream XC dataset.

Candidate corpora details are described in Section A.3.1.

---

[6] `https://storage.googleapis.com/birb-public-bucket/xeno-canto/taxonomy_info_2022-07-18.json`

[7] Refer to `https://storage.googleapis.com/birb-public-bucket/xeno-canto/audio-data/abbbab1/XC116328.mp3`

Benchmark Pipeline Preprocessing

We apply the following preprocessing operations to the evaluation data. An example of this is given in our codebase in `birb/configs/eval_protocol_v2_base.py`.

For species exemplars used to form queries, we apply the same preprocessing to **all** XC focal recordings for a given species in the same manner as for training. Instead of taking random 5s chunks of the 6s audio slices, we keep the middle 5s of the recording (i.e. starting at 0.5s, ending at 5.5s).

For XC-based candidate corpora (in particular, for New York State, USA, Island of Hawai'i, USA, Colombia & Costa Rica) and all soundscape datasets, we apply 5s strided windowing (window length of 5s, window stride of 2.5s), followed by densely annotating each window (i.e. propagating each recording's label to all of the windows produced).

### A.3.2  METRICS

Previous benchmarks and competitions on bird vocalizations have often used mean average precision (MAP) averaged across species (cMAP) to measure performance. Here we provide more detail as to why we believe ROC-AUC is a more appropriate metric in this setting.

ROC-AUC is a scaling of the $U$ statistic from the Mann-Whitney $U$ test (Mason & Graham, 2002), i.e. it measures how well positives are ranked above negatives. The bpref (Buckley & Voorhees, 2004) metric is also a scaled version of ROC-AUC. This metric was found to correlate better with human judgements in retrieval tasks, particularly when there are few labeled examples. From the definition of bpref we can see that ROC-AUC can be viewed a generalization of the normalized rank, similarly to how average precision can be viewed as a generalization of the reciprocal rank. An interpretation of ROC-AUC that follows is that one minus the ROC-AUC score is the fraction of the dataset that will be incorrectly ranked higher than the relevant recording, on average. Fuhr (2018, Sections 2.1 and 2.2) argues that reciprocal rank metrics should be avoided because they lead to counter-intuitive results when averaged, further supporting our choice for ROC-AUC.

### A.3.3  MODELS & TRAINING DETAILS

Classifier models are trained for 1M steps on segments of the upstream XC dataset selected by a peak-finding algorithm. The model input audio is initially transformed to a PCEN Mel-Spectrogram (Wang et al., 2017), with a 100Hz frame-rate.[8] Models are trained with a binary cross-entropy loss, treating every output as an independent binary classifier. We perform a hyperparameter sweep (refer to Table 4 for hyperparameter values) using the cROC-AUC on the Pennsylvania, USA dataset as our validation metric.

When enabled, data is augmented during training with simplified MixUp (Zhang et al., 2017), random peak-normalization, and random temporal cropping. When enabled, the taxonomic auxiliary loss has the model predict the higher levels of the bird taxonomy using additional classification heads.

A full implementation of the network architectures can be found in the accompanying code:

- **EfficientNet** is implemented in the `birb.models.efficientnet` module (refer to the `get_model_config` function in the `small_efficientnet_train_and_valid` and `large_efficientnet_train_and_valid` modules of the `birb.configs.baselines` package for the configuration of the small and large variants, respectively).

- **Conformer** is implemented in the `birb.models.conformer` module (see the `get_model_config` function in the `small_mel_conformer_train_and_valid` and `large_mel_conformer_train_and_valid` modules of the `birb.configs.baselines` package for the configuration of the small and large variants, respectively).

---

[8] A complete implementation can be found in the `get_pcen_melspec_config` function in the `birb.configs.baselines.presets` module of the accompanying code.

In addition to the embedding models introduced in Section 4.1, we include results on two additional models. AudioMAE (fine-tuned) is trained in the same manner as the AudioMAE model with additional supervised fine-tuning on AudioSet labels. S EfficientNet (all XC trained) is presented with the entire XC corpus including Aritifically Rare species in abundance and the species from the heldout Island of Hawai'i, USA and Colombia & Costa Rica.

Hardware & library versions

The BIRB codebase is implemented in Python v3.10 and JAX v0.4.13; we support Python v3.10+ and JAX v0.4.12+. The datasets are implemented using TensorFlow Datasets library. A full list of dependencies and library versions is available in the `pyproject.toml` file of the accompanying code.

Baseline training and model evaluation is performed on Cloud TPU v2 devices with a $2 \times 2 \times 1$ topology.

## A.4 ADDITIONAL LARGE-SCALE INVESTIGATION & DISCUSSION

We provide a series of ablation studies and discussion, followed by additional empirical results on cROC-AUC for each evaluation region and ROC-AUC results on a per-species basis.

### A.4.1 ABLATION STUDY RESULTS

The transfer from focal to passive recording settings required by our benchmark is a complex and multi-faceted problem: it results in extreme covariate shift—due to the differences in recording settings—and label shift—as a result of narrowing the scope from thousands of species distributed globally down to tens or hundreds of species concentrated in a specific geographical location. We characterize the learning challenges presented by our benchmark through ablation experiments disentangling how they individually contribute to the observed reduction in performance on soundscape datasets. In this study, we use Xeno-Canto data for training and evaluation (described below) in addition to Pennsylvania, USA as a validation dataset (also used during embedding model validation in large-scale empirical analysis, Section 4). Note that all Pennsylvania, USA species are contained within XC.

We construct splits of the full Xeno-Canto dataset into data for training and two evaluation datasets, (1) an $i.i.d.$ evaluation dataset which shares the training dataset's class distribution, and (2) a *label-shifted* evaluation dataset which shares Pennsylvania, USA's class distribution. We use Pennsylvania, USA for these ablations to align with the validation soundscape dataset used for validation on our baseline XC-trained embedding models. We also sample a small $i.i.d.$ (same class distribution) subset of the training dataset to compute training metrics on-the-fly.

With the above splits, we can then investigate the isolated effect of label shift by comparing the evaluation performance on the $i.i.d.$ eval dataset and on the label-shifted eval dataset. There is no covariate shift in this case since training and evaluation are both performed using Xeno-Canto data; any difference in evaluation-time performance can be attributed to label shift. Furthermore, we can investigate the isolated effect of covariate shift by comparing the performance on the label-shifted

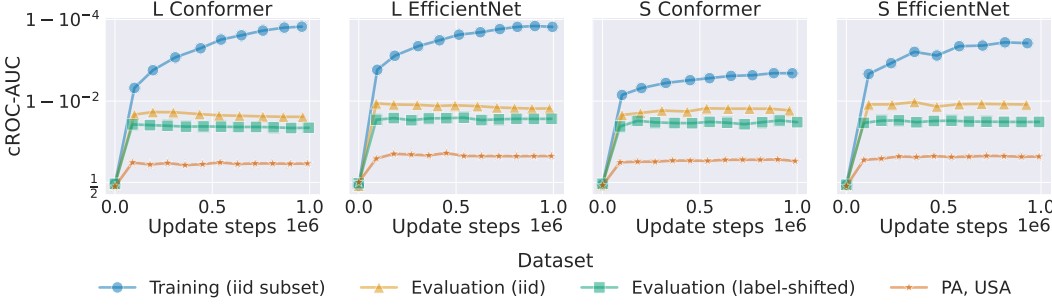

Figure 4: Ablation experiment: cROC-AUC throughout upstream training on the upstream and evaluation datasets.

eval dataset with performance on Pennsylvania, USA. Since, by construction, they have the same label distribution, they are subject to the same label shift with respect to the training set, and any difference in performance can be attributed to the covariate shift from Xeno-Canto to Pennsylvania, USA.

We trained each of our baselines on the aforementioned training dataset and measured ROC-AUC on the various datasets described above throughout training (Figure 4). The effect of label shift is represented by the gap between the *evaluation i.i.d.*) (orange triangle) and *evaluation (label-shifted)* (green square) curves, and the effect of covariate shift is represented by the gap between the *evaluation (label-shifted)* (green square) and Pennsylvania, USA (red star) curves. The usual iid generalization gap can be measured by comparing the *training* ($i.i.d.$ subset) (blue circle) and *evaluation* ($i.i.d.$) (orange triangle) curves. Relatively speaking, we observe that the effect of covariate shift is greater than that of label shift. However, taking a step back, we conclude that the generalization challenge presented by the soundscapes transfer problem is multi-faceted, as we observe generalization gaps that can be attributed to many different causes.

Note that these ablations are not directly capturing the effect of class imbalance. In fact, we observe a non-trivial "classical" generalization gap between *training* ($i.i.d.$ subset) (blue circle) and *evaluation* ($i.i.d.$) (orange triangle), which suggests some amount of overfitting. However, a further break-down of these results per species (as shown in e.g. Figure 5) reveals that this gap varies largely across species, which may be explained by the enormous class imbalance (resulting in more "classical" overfitting in the case of rare species).

In Figure 5, we present model performance as a function of learning for different datasets (evaluation, label-shifted evaluation, validation, and training $i.i.d.$ subset) on a small selection of species. The plots illustrate the diversity of learning challenges that becomes apparent when isolating training performance by species.

We enumerate the following observations for each of the species in Figure 5:

- Performance on the **American Goldfinch** (Figure 5a) doesn't appear to be significantly impacted by label shift (i.e. minimal difference between $i.i.d.$ datasets as eval (label-shifted)); covariate shift appears to be the dominant generalization challenge as evidenced by performance on Pennsylvania, USA.

- Performance on the **Black-and-white Warbler** (Figure 5b) suffers significantly in the presence of label shift (i.e. training and eval ($i.i.d.$) compared to eval (label-shifted)); performance also drops as a result of covariate shift (i.e. $i.i.d.$ sets compared to Pennsylvania, USA).

- Performance on the **Black-capped Chickadee** (Figure 5c) is limited by non-$i.i.d.$ generalization difficulties (i.e. comparing $i.i.d.$ data performance with eval (label-shifted)) and covariate shift (i.e. in light of Pennsylvania, USA performance relative to other datasets).

- The models generalize well for the **Chestnut-sided Warbler** (Figure 5d), even in the presence of label shift and covariate shift.

- Performance on the **Scarlet Tanager** (Figure 5e) is limited by the challenges of label and covariate shift.

### A.4.2 MODEL PERFORMANCE BY REGION

We present cROC-AUC metrics broken down by evaluation region, corpus, and model in Figure 6 and Figure 7a.

Recall that we process XC recordings by extracting audio slices using a peak-finding algorithm (Appendix A.3.1), and the recordings' foreground label is assigned to each peak-found slice. XC recordings also feature optional background labels; when present, we assign them to each peak-found slice as well. For a given region, the candidate corpus contains all of the windowed recordings in which the region's species appear. In particular, for XC Focal corpora, the windows for candidate species within a region with **foreground** labels are included (where we explicitly exclude windows with background vocalizations); similarly, windows with **background** labels comprise its XC Background corpus (where focal windows are excluded).

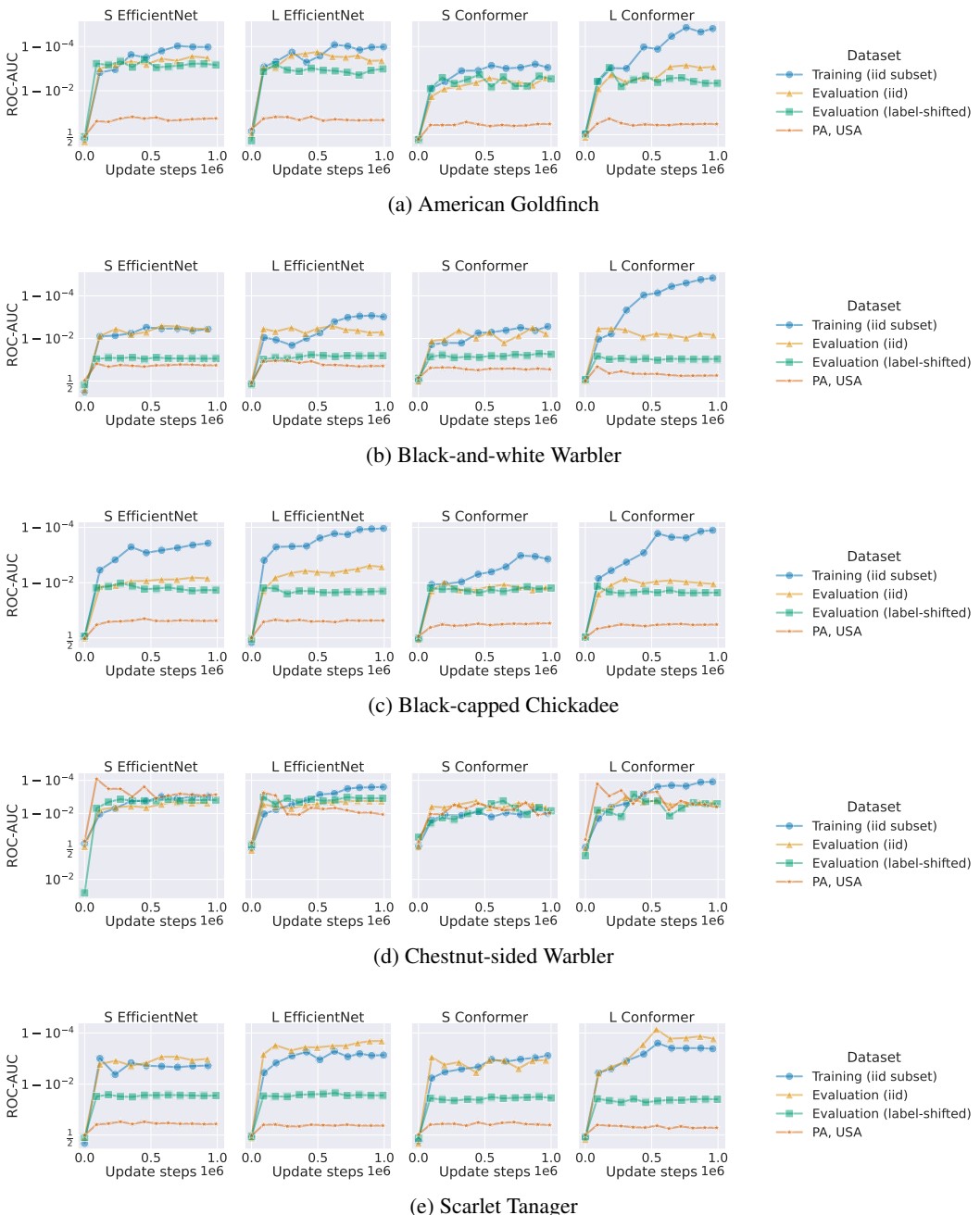

Figure 5: ROC-AUCs (in logit scale) on select species throughout training on the training XC ablation dataset.

It's important to acknowledge that the background labels within XC are significantly noisier than foreground labels, in part because they are optional and may be missing from recordings in some cases. Background labeling can be more challenging for annotators due to background noise, overlapping calls, etc. Furthermore, there are differences in label fidelity between the XC and Soundscape datasets for each region. In particular, the Soundscape datasets provide the highest label quality given the rigorous annotation procedure that experts follow. XC Focal labels have the next highest degree of label confidence, but exhibit more noise relative to Soundscapes because annotation is crowdsourced from contributors with varying levels of expertise. Due to the aforementioned challenges, XC Background presents the most label uncertainty of the three dataset types. In addi-

tion to label noise, there is a covariate shift between XC Focal and Background corpora and their Soundscape counterpart.

In Section 4, we provide thorough discussion comparing XC Focal and Soundscape performance for all benchmark tasks. We include performance on XC Background corpora for each region in Figure 6.

For the Artificially Rare species (New York State, USA, Figure 6a), it appears that XC Background presents the most generalization challenges relative to XC Focal and Soundscapes; however, due to label noise it's unclear if the observed performance is statistically significantly different with regards to the top performing models for that region. When comparing XC Focal and XC Background, we empirically witness some combination of the effects of covariate shift and difference in label quality between the datasets. Further investigation would be interesting in disentangling the sources of performance difference.

AudioMAE (fine-tuned) compared with AudioMAE does not exhibit a clear trend in performance difference on any of the tasks (Figure 6, Figure 7a). This further motivates the benefit of pre-training models on XC for avian bioacoustic tasks.

Comparing our two S EfficientNet embedding models, we are surprised at the relatively marginal difference in performance when this architecture is given access to the entire XC dataset versus the XC upstream data at training time. In particular, on the Artificially Rare species in New York State, USA, the performance is nearly indistinguishable across the three corpora. The two Heldout regions, Island of Hawai'i, USA and Colombia & Costa Rica, reflect tasks with the most significant discrepancy in training-time data availability; further investigation is necessary to better understand what contributes to the insignificant difference in model performance.

### A.4.3 SOUNDSCAPE DATA AVAILABILITY

When breaking down performance by species availability (Figures 7b and 7c), we observe that Heldout species exhibit high variability in cROC-AUC compared to those available upstream. The reasons for the lower empirical performance across all models on available species vs. Heldout species for the Sierra Nevada, USA region and the inverse for Peru are unclear. Investigation into the qualities of these species' vocalizations, representation in the dataset, and similarity with upstream-available species are interesting areas of follow-up to better understand the variability in model performance on these Soundscapes.

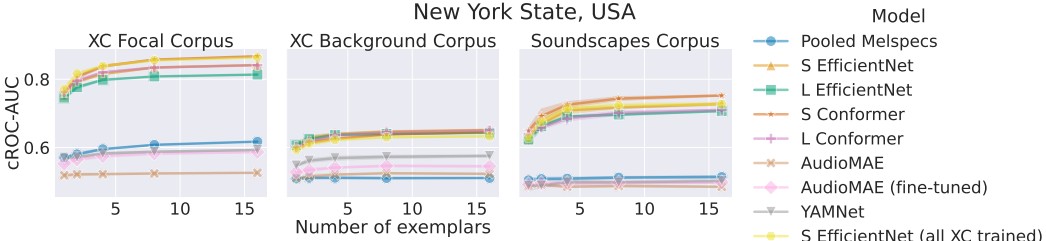

(a) cROC-AUC performance across models on the Artificially Rare New York State, USA set.

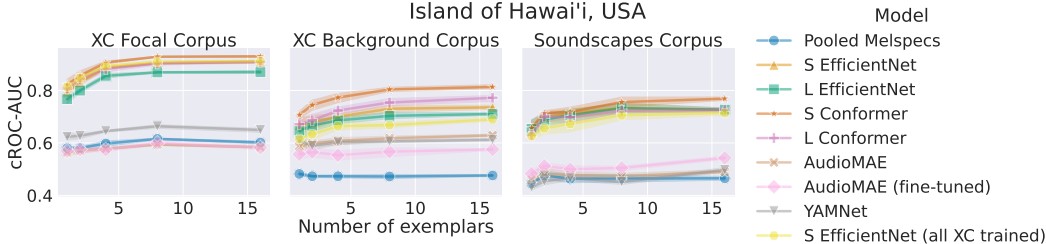

(b) cROC-AUC performance across models on the Island of Hawai'i, USA evaluation set (all classes held out from upstream data).

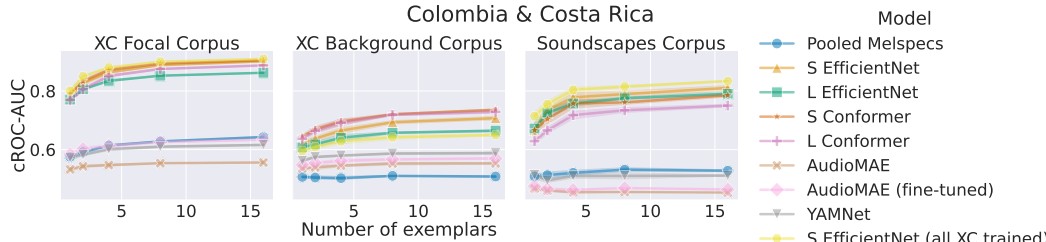

(c) cROC-AUC performance across models on the Colombia & Costa Rica evaluation set (all classes held out from upstream data).

Figure 6: cROC-AUC performance as a function of number of exemplars for each embedding model on Artificially Rare (New York State, USA) and Heldout (Colombia & Costa Rica, Island of Hawai'i, USA) evaluation regions.

Table 4: High-level summary of BIRB baseline model training details. *Note that 6s slices are taken, and then truncated to random 5s subslices.

| Component | Detail | Description |
|---|---|---|
| Class labels | Taxonomy | eBird 2021 |
| | Ignored species | `reevir1, gnwtea, grnjay, butwoo1` |
| Peak-finding | Slice length | 6 seconds* |
| | Maximum number of slices | 5 |
| | Mel-spectrogram | 80 milliseconds window |
| | | 10 milliseconds hop |
| | | logarithmic scale (floor 0.01, rescale 0.1) |
| Baseline training | Parameter updates | 1M |
| | Input representation | PCEN Mel-spectrogram |
| | Hyperparameters | learning rate $\in$ {1e-5, 3.16e-4, 1e-3, 3.16e-1, 1e1} |
| | | learning rate cosine decay $\in$ {on, off} |
| | | Taxonomic auxiliary loss $\in$ {on, off} |
| | | Random augmentations $\in$ {on, off} |
| | Validation dataset | Pennsylvania, USA |

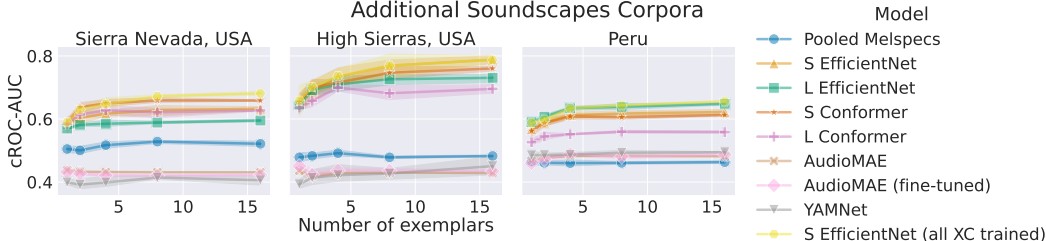

(a) cROC-AUC performance across models on additional soundscapes corpora (Sierra Nevada, USA, High Sierras, USA, Peru).

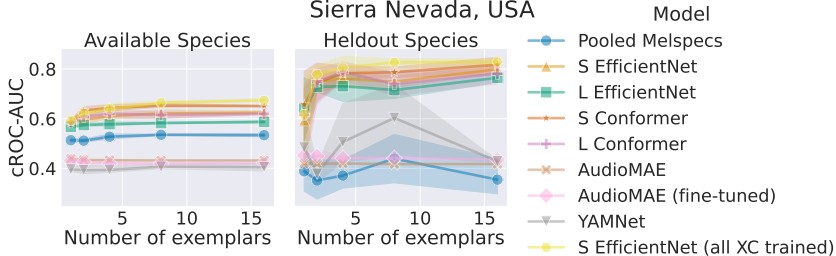

(b) cROC-AUC performance across models on the Sierra Nevada, USA set, by species availability.

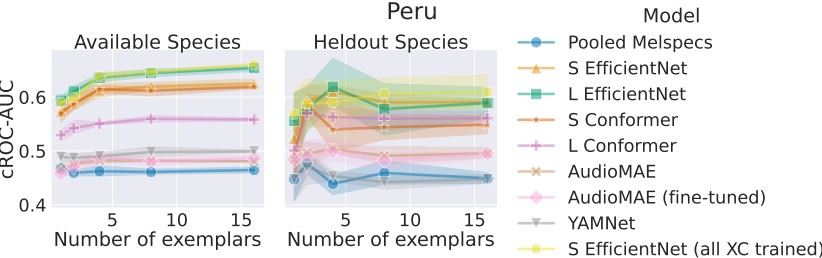

(c) cROC-AUC performance across models on the Peru evaluation set, by species availability.

Figure 7: cROC-AUC performance across models as a function of number of exemplars on Sierra Nevada, USA and Peru evaluation regions. High Sierras, USA species are completely overlapping with the XC upstream dataset.

