# OpenReview forum: "BIRB: A Generalization Benchmark for Information Retrieval in Bioacoustics"
_ICLR.cc/2024/Conference — Submitted to ICLR 2024_

### Official Review · Reviewer_sjxb · 2023-10-18

**Soundness:** 3 good
**Presentation:** 2 fair
**Contribution:** 3 good
**Rating:** 5
**Confidence:** 3

**Summary:**

The paper proposes a benchmark (BIRB) centered on the retrieval of bird vocalizations. It contains multiple passively-recorded datasets and a baseline system for these tasks are proposed. The benchmark aims for the direction of ML models' robustness and generalization ability.

**Strengths:**

1. The paper proposes a new series of datasets, covering various aspects of challenging ML problems, especially in the field of acoustic data shift.
2. A pipeline for performance evaluation is also proposed. The authors find that increasing the size of networks does not translate to improved generalization, which is an interesting phenomenon.
3. Various models are tested and benchmarked on the datasets.

**Weaknesses:**

While the topic of this paper is intriguing, I have some concerns that lowers my score:
1. The paper claimed the finding of "increasing the size of networks does not translate to improved generalization". However, no reason for this phenomenon is given.
2. Many evaluation datasets are proposed. Does each of these datasets emphasize something (e.g. Label shift influence, long-tail distribution)? If not, they lack of research potential to be set as a part of benchmark dataset. A good example for constructing evaluation set would be ImageNet-C[1].
3. Throughout the paper, no example dataset sample is provided to give intuition of how the dataset is constructed.

[1] Benchmarking Neural Network Robustness to Common Corruptions and Perturbations. ICLR 2019

**Questions:**

I also have the following questions:
1. Is it a reasonable practice to use existent public datasets to construct one's own dataset? Are the credits properly given?
2. What does BIRB stand for? Apologies in advance if I missed it in the paper.

**Details Of Ethics Concerns:**

The authors have discussed ethical concerns.

---

> ### Author Response · Authors · 2023-11-16
> **Response to Reviewer sjxb**
>
> We thank Reviwer sjxb for their feedback. We hope the following addresses the weaknesses and questions raised in their review.
>
> > The paper claimed the finding of "increasing the size of networks does not translate to improved generalization". However, no reason for this phenomenon is given.
>
> This is a good question. Prior works have also made similar observations (for instance "A Unified Few-Shot Classification Benchmark to Compare Transfer and Meta Learning Approaches"). Further investigation is necessary, but our intuition is that either a) scaling up capacity is not an effective strategy in challenging transfer tasks, or b) due to class imbalance, we are in a medium-to-low data regime for a significant proportion of the species that is not supportive of larger network architectures.
>
> This finding highlights the importance of our objective of designing a benchmark that is informed by a realistic problem setting. Standard academic benchmarks are akin to unit tests for deep learning approaches: methods developed using those benchmarks can and do overlook important failure cases in real-world applications, which can only be uncovered through "integration test" benchmarks like BIRB.
>
> We will incorporate this discussion in the updated manuscript.
>
> > Does each of these datasets emphasize something (e.g. Label shift influence, long-tail distribution)? If not, they lack of research potential to be set as a part of benchmark dataset.
>
> Each of the datasets maps to one of the tasks we describe in Section 4.3, each of which aligns to one or more challenges that we measure by evaluating on that dataset. While the alignment is not necessarily one-to-one, the datasets are diversified across geographical locations and species distributions, which helps establish generalizability in the benchmark results. We also note that the "Disentangling Label and Covariate Shift" subsection in Section 4.3 does present ablations that directly disentangles the various learning challenges presented by BIRB. The methodology we present there can be followed by users of the benchmark to study the effect of those challenges in isolation.
>
> "Integration" benchmarks like BIRB are important to deep learning researchers for the reasons outlined in our response to Weakness 1. Our benchmark is intended to drive fundamental research in a way that is directly beneficial to ecological researchers and conservationists, whose modelling challenges do not neatly decompose into different research subfields. Transfer learning, domain adaptation/generalization, and class-imbalanced learning applications all have the opportunity to demonstrate effectiveness on BIRB, and we believe that the fact that such advances can directly translate to real-world impact will be encouraging to many researchers in those fields.
>
> > Throughout the paper, no example dataset sample is provided to give intuition of how the dataset is constructed.
>
> To clarify, are you referring to an example as it goes through the data processing pipeline, or do you mean more explicitly the modality/format of an example? We'd be glad to provide a more explicit description and/or additional modality to illustrate this more clearly--please let us know which component you were referring to.

---

> > ### Comment · Reviewer_sjxb · 2023-11-17
> > **Comment**
> >
> > Thank the authors for their detailed responses and I really appreciated it. The response has fully resolved my concerns in:
> > 1. The usage (intention) of the constructed test sets.
> > 2. The validity of the proposed dataset.
> > 3. The full name for BIRB.
> >
> > For the third weakness, I meant the modality/format of an example. Sorry for the ambiguity.
> >
> > The only left concern is the presentation of the paper (also mentioned by other reviewers), and could be improved from the following aspects (from my opinion):
> > 1. Careful claims: Every conclusion should be well supported explicitly, especially with the prerequisite "we find".
> > 2. Better introduction to the research topic.
> > 3. A brief guide for benchmark usage. Since one of the paper's goal is to encourage and inspire machine learning & deep learning researchers to look into the field of bioacoustics, this would be important.
> >
> > Given the current version of the paper, I am keeping my score for now. However, I believe this research work could be impactful and I encourage the authors to revise the paper as much as they can.

---

> > > ### Author Response · Authors · 2023-11-20
> > > **Following-up**
> > >
> > > Dear Reviewer sjxb: Thank you for responding and clarifying!
> > >
> > > We are incorporating and addressing your last concern regarding the presentation of the paper.
> > >
> > > 1. Below we summarize the changes we’ve made and will upload to OpenReview. We would appreciate you sharing if there’s any significant additional change that you would like to see before the camera-ready version if you think it would improve the readability/comprehension/presentation of this work. We will gladly fine-tune our camera ready to make it as useful to the reader as possible. A few changes to highlight:
> > > * We’ve included expansions of the benchmark name (BIRB) in the Abstract and Introduction.
> > > * We’ve incorporated careful discussion in our conclusion anchored on our observations and analysis from our empirical evaluation, which we’ve included below for your consideration:
> > >     * “Based on our investigation, we find that the shift from focal to passive recordings contributes substantial generalizability challenges for embedding models, surprisingly even when increasing model size by a significant factor. Beyond covariate shift, our empirical results indicate that increasing the size of architectures we evaluate does not provide improved generalization in terms of label shift, robustness to distribution shift, and performance on novel species. While further study is needed, our intuition is that despite being fairly large, the \XC dataset we train on is highly class-imbalanced and a significant portion of the bird species (classes) reflect a medium-to-low data regime that is not conducive to larger network architectures.”
> > > * To provide a clearer high-level overview of the baseline system/protocol, we are moving the original figure to the appendix and are replacing it with an updated and simplified summarization figure that includes the following pieces:
> > >     * A “cartoon” image of an audio slice converted into spectrogram format (sound to image representation) to clarify the input data format
> > >     * A query and a sliced search corpus (both as spectrograms) as inputs to the embedding model
> > >     * Top m matches from the search corpus (likely as a ranking of search corpus slices sorted by similarity score) as output of the baseline system.
> > >
> > > 2. By “introduction to the research topic”, do you mean specifically into ML x bioacoustics? If so, can you please clarify what pieces would better align a reader from an ML background to this work? We hope that the figure sketch we described in the point helps clarify the baseline system. If you have any concrete suggestions, we appreciate you elaborating so we can better understand how we might improve our introduction to our research problem.
> > >
> > > 3. To your point on a guide for benchmark usage: we’ve already opensourced the end-to-end benchmark on GitHub (refer to the Supplementary Material for an anonymized mirror of the opensourced codebase). With this, we provide the baseline setting with clear documentation for the user to extend this to their choices (such as their input data, embedding model, similarity metric, evaluation metrics, etc.). This supports adoption of this benchmark for ML & DL researchers to extend our system to investigate their advancements in related approaches for bioacoustics. Furthermore, we include an elaborated introduction to [avian] bioacoustics and challenges (Appendix A.1) in this space to inspire researchers and developers to inform their research and ML innovations.
> > >
> > > We are grateful for your engagement! Please let us know if there are other aspects of the presentation that you think are worth adjusting.

---

> ### Author Response · Authors · 2023-11-16
> **Response to Reviewer sjxb**
>
> Regarding the two questions raised in the review:
>
> > Is it a reasonable practice to use existent public datasets to construct one's own dataset? Are the credits properly given?
>
> There is precedent in constructing "datasets of datasets" benchmarks; refer for instance to "A Large-scale Study of Representation Learning with the Visual Task Adaptation Benchmark" or "Meta-Dataset: A Dataset of Datasets for Learning to Learn from Few Examples". We agree that proper credit attribution is important in those cases, which we do through citations in Table 1. We will clarify in the text that we are making use of data collected as part of previous works to help avoid any ambiguity, and we are happy to incorporate any further suggestion on how to make that clear.
>
> We also note that the datasets we employ are designed for ecologically themed research and are not usable off-the-shelf in machine learning applications. They do not prescribe any specific evaluation procedure; the evaluation protocol we propose is our own contribution. The datasets were also prepared by a variety of different groups, which required significant effort on our part to achieve data consistency across datasets. Preprocessing details (for reproducibility) are outlined in Appendix 1, which we summarize here:
>
> - Resolving significant differences in the bird taxonomies used by various datasets
> - Correctly and consistently aligning labels in various input formats
> - Extracting fixed length slices from file-level labels/annotations using peak-finding (Xeno-Canto queries)
> - Converting timeboxed annotations in soundscape data into fixed length labeled slices
>
> > What does BIRB stand for? Apologies in advance if I missed it in the paper.
>
> Thank you for pointing this out. We will clarify by expanding the acronym (Benchmark for Information Retrieval in Bioacoustics) the first time it is used in the main text. We will also add a footnote to an online article (https://www.audubon.org/news/when-bird-birb-extremely-important-guide) to provide context on the origin of the word.

---

### Official Review · Reviewer_sG4k · 2023-10-31

**Soundness:** 2 fair
**Presentation:** 2 fair
**Contribution:** 3 good
**Rating:** 5
**Confidence:** 3

**Summary:**

This paper is benchmarking a much more complicated evaluation scenario for modern machine learning algorithm. Specifically, it designs a information retrieval task based on bird vocalizations. Several existing public datasets are involved to create this benchmark. Accordingly, the author also provide several baselines for this benchmark. The machine learning generalization ability and robustness property are expected to be evaluated based on this real-world fashion benchmark.

**Strengths:**

1. The research topic is attractive to me. Considering a real practical scenario for current machine learning field is reasonable and necessary.

2. The collected datasets are sufficient and provided baselines cover some recent popular methods.

**Weaknesses:**

I mainly concern the presentation of this paper. Given the current machine learning field not familiar with the bird-based bioacustics, more background information should be included before introducing the benchmark. Similarly, the benchmark itself is also unclear. The task should be better to illustrate with a figure and the figure of baseline system is not informative. Some relevant information in supplementary may be moved into the main draft to elaborate the benchmark. In addition, for the experimental analysis, corresponding discussion and visualizations are necessary for a better description.

**Questions:**

Please refer to the weakness for my concerns and most of the paper format should be improved for a better illustration to readers. I recognize the research contribution of this paper but current draft is a bit unclear for a good publication. The author may want to significantly revise the paper draft to clarify this benchmark work.

---

> ### Author Response · Authors · 2023-11-15
> **Response to Reviewer sG4k**
>
> We thank Reviewer sG4k for their feedback! We address some concerns and include follow-up questions so we can better understand the concerns.
>
> > Similarly, the benchmark itself is also unclear. The task should be better to illustrate with a figure and the figure of baseline system is not informative.
>
> Would you be able to clarify the aspects which are unclear so we can address these either with clarifications in our response or improve the presentation of the paper? We care about accessibility of this system and reader understanding!
>
> > Some relevant information in supplementary may be moved into the main draft to elaborate the benchmark. In addition, for the experimental analysis, corresponding discussion and visualizations are necessary for a better description.
>
> Are there any pieces included in the supplementary that you found particularly helpful to highlight in the main paper? For the sake of providing sufficient introduction to the setting, the dataset descriptions, benchmark setup and protocol details, suite of evaluation results and analyses, etc., there is a lot of content and discussion to cover and we’re trying to strike a balance between providing enough information in the main paper and opportunities for the reader to learn more by referencing the Appendix where we elaborate on all of these areas.
>
> > Given the current machine learning field not familiar with the bird-based bioacustics, more background information should be included before introducing the benchmark.
>
> Similarly to the previous point, if you have any specific suggestions on what you found or would find helpful, please do let us know!

---

> > ### Author Response · Authors · 2023-11-20
> > **Follow-up**
> >
> > Dear Reviewer sG4k: would you please clarify the points you found unclear? We’ve aimed to address your concerns, but without response from you it is hard to take your feedback into account. Based on your comment about our baseline system figure (Figure 1) and another reviewer’s comment about data format, we are moving the original figure to the appendix and are replacing it with an updated and simplified summarizing figure that will include the following pieces:
> >
> > * A “cartoon” image of an audio slice converted into spectrogram format (sound to image representation) to clarify the input data format
> > * A query and a sliced search corpus (both as spectrograms) as inputs to the embedding model
> > * Top m matches from the search corpus (likely as a ranking of search corpus slices sorted by similarity score) as output of the baseline system.
> >
> > Does this clarify the basic setting for you? We await your reply on the prior points--thank you.

---

### Official Review · Reviewer_N4MM · 2023-10-31

**Soundness:** 2 fair
**Presentation:** 2 fair
**Contribution:** 3 good
**Rating:** 5
**Confidence:** 2

**Summary:**

The paper introduces BIRB, a benchmark for bioacoustics, addressing challenges in machine learning generalization. BIRB focuses on a retrieval task where models are trained on an upstream dataset and tested on retrieving vocalizations from a different corpus. It assesses out-of-distribution generalization, few-shot learning, and robustness to class imbalances. The benchmark provides a baseline system using a nearest-neighbor search for efficient evaluation. BIRB has practical implications for bioacoustics research and large-scale data processing, offering a real-world and complex evaluation platform for machine learning models in bioacoustics.

**Strengths:**

1. The paper introduces the BIRB benchmark, which is a novel and comprehensive benchmark designed for evaluating model generalization in the field of bioacoustics. The paper's originality lies in its approach to formulating retrieval tasks for bird vocalizations, capturing real-world complexities, and addressing generalization challenges specific to this domain.
2. Moreover, this work's strength lies in its use of publicly available, high-quality bioacoustic datasets, such as Xeno-Canto and passively collected soundscape datasets.
3. This paper's primary contribution is in the proposed benchmark, which opens up opportunities for researchers to explore and innovate in the field of bioacoustics. While the proposed method (baseline) is not the central focus, it provides a practical starting point for conducting retrieval tasks within the benchmark.

**Weaknesses:**

1. While the paper primarily focuses on introducing the benchmark, it could benefit from more innovative approaches or methodologies for retrieval tasks. I think it's important to inspire researchers with novel ideas for addressing the challenges in bioacoustics, beyond providing a baseline model.
2. The paper could be enhanced by a more extensive comparative analysis of different approaches or models for the tasks presented in the benchmark. Currently, it provides results for a set of baseline models but does not explore alternative methodologies.
3. This work provides detailed results but could improve the interpretation of these results. For instance, it mentions that there's a significant difference in performance between deep models trained on XC upstream data and models pre-trained on AudioSet, but it doesn't delve into the reasons behind this discrepancy or suggest potential solutions.
4. The paper employs ROC-AUC as its primary evaluation metric. While this is common in information retrieval tasks, it would be beneficial to consider additional evaluation metrics that are specific to bioacoustics and relevant to the benchmark's objectives.

**Questions:**

1.  The paper mentions performance differences between models pre-trained on AudioSet and those trained on XC upstream data. Could the authors offer more insights into why this divergence occurs and what it might imply for domain adaptation in bioacoustics?
2. Are there any specific experiments, ablations, or investigations that the authors plan to conduct in the future with the benchmark, beyond the preliminary baseline models presented in this paper?
3. Are there specific implications for domain adaptation, representation learning, or transfer learning from the challenges in bioacoustics and the proposed benchmark?

---

> ### Author Response · Authors · 2023-11-15
> **Response to Reviewer N4MM**
>
> We thank Reviewer N4MM for their thoughtful feedback and questions! We address these concerns and questions.
>
> > [The benchmark] could benefit from more innovative approaches or methodologies for retrieval tasks. I think it's important to inspire researchers with novel ideas for addressing the challenges in bioacoustics, beyond providing a baseline model.
>
> The ICLR 2024 call for papers explicitly mentions datasets and benchmarks in the list of relevant topics. As such, our contributions in designing an evaluation protocol which closely aligns with the real-world needs of ecological researchers and conservationists and demonstrating that it represents a significant and multi-faceted challenge stand on their own. In the evaluation framework we present, we encapsulate naturally occurring challenges, align these with important ML problem areas, and disentangle the influences of the underlying challenges into a comprehensive suite of tasks. Furthermore, in our initial empirical investigation and evaluation, we characterize the extent to which each provides challenges for the models we benchmark. Lastly, we provide inspiration to researchers by outlining concrete follow-up directions by enumerating our findings and open problems that stem from these.
>
> > The paper could be enhanced by a more extensive comparative analysis of different approaches or models for the tasks presented in the benchmark.
>
> Within the extensive comparative analysis that we've presented in this work, are there any specific approaches or models that you think would be important to include in providing this first set of comprehensive evaluations, results, and analyses?
>
> > In regards to Weaknesses 3 and Question 1:
>
> A meaningful insight from our work reveals the importance of using an appropriate learned representation in this framework when addressing bioacoustics retrieval. In particular, the model trained on XC upstream is more relevant and applicable to this task, providing a rich feature space that translates well to downstream tasks, much more so than the AudioSet dataset which includes many unrelated or irrelevant acoustics data.
> For domain adaptation, our conclusion is that we benefit from training on data that is more directly related than data that is less so. More explicitly, bird vocalizations provide a more useful representation for detecting other bird vocalizations than the broad and less relevant classes included in AudioSet.
>
> > [It] would be beneficial to consider additional evaluation metrics that are specific to bioacoustics and relevant to the benchmark's objectives.
>
> ROC AUC is also common for bioacoustics tasks; refer for instance to Stowell et al. (2022): "Performance is measured using standard metrics such as accuracy, precision, recall, F-score, and/or area under the curve (AUC or AUROC)". Appendix A.3.2 provides a justification for our choice of metric. We will include a statement in the main paper to highlight this point.
>
> > Are there any specific experiments, ablations, or investigations that the authors plan to conduct in the future with the benchmark, beyond the preliminary baseline models presented in this paper?
>
> As we touched on in response to your concerns in Weaknesses #1, our thorough investigation has provided meaningful open problems (in the context of foundational ML research and bioacoustics-related fields) and future work within this setting, as we have outlined in Section 6: Conclusion. Our group continues to explore problems within this space and are particularly interested in building self-supervised approaches that leverage large quantities of unlabeled and in-domain data.
>
> > Are there specific implications for domain adaptation, representation learning, or transfer learning from the challenges in bioacoustics and the proposed benchmark?
>
> Implications include:
> 1. Using a relevant and general dataset for learning a rich and transferable feature space is a powerful approach. This is apparent from our comparison between XC models and AudioSet pre-trained models.
> 2. There is value in combining the baseline approaches we presented with domain adaptation techniques in building robust and generalizable retrieval models. This motivates the continued exploration of the intersection between domain adaptation, transfer learning, and bioacoustics, given that bioacoustics provides an inherently more complex backdrop than many academic domain adaptation settings.
> 3. As we previously stated, bioacoustics represents a rich playground to address fundamental ML questions but is underserved when focusing only on the canonical settings that are often explored, i.e. computer vision and language tasks.

---

> > ### Author Response · Authors · 2023-11-20
> > **Follow-up**
> >
> > Dear Reviewer N4MM: we would value your input on the approach(es) you think are important to compare against within the presentation and baseline evaluation of the BIRB benchmark.
> >
> > It’s worth acknowledging that extending beyond the baseline setting we present in this work is itself a viable standalone research problem and contribution. Adopting a “more complex” approach requires either applying it as an “off-the-shelf” approach and accepting that it may perform quite poorly on some of these tasks (which isn’t necessarily conclusive of the approaches applicability for this setting or tasks), or requires substantial innovation to address the complex challenges included in this benchmark, i.e. audio domain (not image), substantial class imbalance, multiple sources of distribution shift.
> >
> > Beyond that point, we believe we addressed your other questions and concerns; if you agree, would you be willing to update your score?

---

> > > ### Comment · Reviewer_N4MM · 2023-11-22
> > > **Response to the Authors**
> > >
> > > Thank you for your responses. I appreciate your efforts to address my comments and questions. However, I still have some reservations about your paper, especially regarding the novelty and significance of your approaches and methodologies. I think your paper would benefit from more innovative and inspiring ideas for tackling the challenges in bioacoustics, beyond providing a benchmark and some baseline models. Therefore, I am not convinced to increase my score at this point. I hope you can accept my feedback and improve your paper in the future.

---

> > > > ### Author Response · Authors · 2023-11-22
> > > > **Clarifications**
> > > >
> > > > Thank you for responding. We disagree with the claim that our submission should present innovative or significant approaches on top of the initial baselines we contribute in this work (in addition to the framework). The [ICLR 2024 Call for Papers](https://iclr.cc/Conferences/2024/CallForPapers#:~:text=datasets%20and%20benchmarks) explicitly lists datasets and benchmarks as part of the relevant topics. A reviewer insisting that our submission should also present novel technical contributions on top our our extensive benchmarking contributions (from your review: "a novel and comprehensive benchmark designed for evaluating model generalization in the field of bioacoustics.") imposes an unfair double burden on us. For your reference, here is a (non-exhaustive) list of benchmark papers ICLR 2023 that demonstrate precedence in accepting work on benchmarking contributions alone:
> > > >
> > > > - [Dr. Spider](https://openreview.net/forum?id=Wc5bmZZU9cy): *Top-5% paper*, Section 4 lists 7 approaches, all prior works, which are evaluated on their benchmark.
> > > > - [DaxBench](https://openreview.net/forum?id=1NAzMofMnWl): *Top-5% paper*, Sections 4.2 through 4.4 list 6 approaches in total, all prior works, which are evaluated on their benchmark.
> > > > - [A framework for benchmarking class-out-of-distribution detection and its application to ImageNet](https://openreview.net/forum?id=Iuubb9W6Jtk). *Top-25% paper*, benchmarks 525 ImageNet-1k classifiers available from popular repositories.
> > > > - [STREET](https://openreview.net/forum?id=1C_kSW1-k0): *Top-25% paper*, all approaches listed in Section 3 either adopt the methodology of or make direct use of prior works.
> > > > - [MEDFAIR](https://openreview.net/forum?id=6ve2CkeQe5S): *Top-25% paper*, Section 3.2 lists 11 algorithms, all prior works, used for evaluation in the benchmark.
> > > >
> > > > We hope this clarifies for you that the benchmarking contributions you noted as part of the submission's strengths do meet the bar for acceptance.

---

### Official Review · Reviewer_SRto · 2023-11-03

**Soundness:** 1 poor
**Presentation:** 2 fair
**Contribution:** 2 fair
**Rating:** 5
**Confidence:** 3

**Summary:**

This paper proposes a benchmark to measure the generalization capabilities of bird vocalization detection models.
The benchmark is composed of existing datasets in the field, one large-scale upstream (i.e. training) dataset and 7 small-scale downstream (evaluation) datasets, which overall evaluate generalization against several challenges, such as domain shift, label shift, limited data and class imbalance.
Models are trained on the upstream dataset and used as "embedding models" (i.e. feature extractors) to solve retrieval tasks using a few labeled instances on the downstream datasets.
7 recent baseline models are evaluated on this benchmark, including linear models trained on handcrafted features and deep embedding models with EfficientNet, Conformer and Transformer architectures.
Results show that domain-shift is one of the most challenging generalization factor.

**Strengths:**

- The benchmark is a nice real-world application on measuring generalization in the field of bioacoustics, which could be interesting for practitioners.
- Evaluation protocol is simple to follow/apply, i.e. models are used as feature extractors and the task is simple retrieval.
- A decent number of baselines already evaluated in this setting.

**Weaknesses:**

## Concerns on the evaluation protocol
- The evaluation benchmark retrieves relevant samples from a downstream dataset given a query, and it allows reusing class embeddings for queries learned during training. I'm not sure to get this point. If the label of a query is known, then what is the point of retrieving samples from a dataset? We already know the answer.
- Regardless, it would be nice to report results where none of the class embeddings from training time is used when evaluating the models. Because, one of the generalization aspects posed by the datasets is domain-shift, and it is not clear how to measure domain-shift when query embeddings come from training dataset (rather then being embedded at test time using downstream dataset images).

## Comments on datasets
- I wonder if there is a universal class taxonomy which encapsulates all the classes in all the datasets. For instance, in standard vision datasets (like ImageNet, MS-COCO), classes come from different ontologies/granularities and matching those classes is far from being trivial. To measure "label shift", it is essential to know which classes have been seen during training vs at test time (i.e. seen and unseen classes).
- Also, when measuring label shift, what is the semantic relation (due to their granularity) between seen and unseen bird classes?
- Is it possible to evaluate domain-shift while fixing label-shift? For instance, seen and unseen classes being equal, whereas recordings being focal vs passive.

## Comments on the paper overall
- Given that this is a benchmark paper, and that not everybody is super familiar with the domain (bioacoustics), I would expect a more direct and clear explanation of the task being solved, the types of input given to the network, etc. In that sense the paper is not easy to understand.

**Questions:**

I would like the authors to address the weaknesses I listed above.
My main concerns are related to soundness of the evaluation protocol.

---

> ### Author Response · Authors · 2023-11-15
> **Response to Reviewer SRto**
>
> We thank Reviewer SRto for their thoughtful feedback! We address the Reviewer’s questions, concerns, and feedback below.
>
> > If the label of a query is known, then what is the point of retrieving samples from a dataset?
>
> The term "query" is used in the "retrieval" sense of the word: the query is not the example we wish to label, but rather the example (or handful of examples) which is a representative of a species’ vocalization that we seek to retrieve in the unlabeled corpus. This represents a practical use case where a practitioner might say: "I have an example of a Canadian goose vocalization (query) and want to find other examples of this species/vocalization type in my unlabeled dataset/field recording (search corpus).”
>
> > It would be nice to report results where none of the class embeddings from training time is used when evaluating the models.
>
> Can you clarify what you mean by class embeddings from training time?
> If we understand correctly, here is some clarification. No examples (audio) used in evaluation are included at training time. This includes both the query examples and search corpora for each dataset/task. Furthermore, we exclude entire subsets of classes from being presented at training time.
>
> > I wonder if there is a universal class taxonomy which encapsulates all the classes in all the datasets.
>
> In the empirical evaluation presented in this work, we produce a "standard" class taxonomy that encapsulates all the classes across every dataset we evaluate on, which is analogous to what you suggest and describe. It is worth noting that there is no universal taxonomy used across the field and that numerous class taxonomies exist and are updated every 6 or 12 months. Additional details on the topic of class taxonomies are included in the section titled "Labels & Species Codes" in Appendix A.3.1.
>
> > To measure "label shift", it is essential to know which classes have been seen during training vs at test time (i.e. seen and unseen classes).
>
> We align with Moreno-Torres et al. (2012)'s definition of label shift (or prior probability shift): a problem for which P(Y) changes between the training and test datasets but P(X | Y) remains the same. In the experiment on disentangling label and covariate shift all classes are seen during training, but their marginal probabilities change in moving from the training set to the "evaluation (label-shifted)" set. In that setting there are no classes that are unseen at training time.
>
> > When measuring label shift, what is the semantic relation (due to their granularity) between seen and unseen bird classes?
>
> Is your question in regards to a specific setting that we evaluate, as in Section 4.3 Generalization Results?
> Overall, the consideration of semantic relation is irrelevant/independent when measuring label shift. It is, however, relevant when considering performance on low resource classes (those with minimal representation at training), as in the case of the Artificially Rare task and matters when considering generalization to such classes. Additionally,
> 1. The ablation study includes all classes observed ("seen") at training time and investigates the impacts of label shift.
> 2. There is a component of label shift in the Artificially Rare task, and all bird classes within that collection are included at training time.
> 3. In the "Performance in mixed conditions" task, there is a combination of bird classes included at training time and those which are not present due to overlap with the Heldout Region sets.
>
> > Is it possible to evaluate domain-shift while fixing label-shift?
>
> Absolutely! Refer to Figure 3: the "evaluation (label-shifted)" dataset is drawn from the same XC dataset as the training set, and therefore is not subject to domain shift, while the "Pennsylvania, USA" dataset is subject to domain shift (focal to passive). The "evaluation (label-shifted)" dataset's marginal class distribution is constructed specifically so as to match the marginal class distribution of the Pennsylvania dataset.
> Comparing the two allows us to assess the residual effect of domain shift while controlling for label shift.
> Note that it's not feasible to instead subsample the Pennsylvania dataset so that its marginal class distribution matches that of XC for two reasons: 1) some classes have a zero marginal probability in the Pennsylvania dataset, and 2) the Pennsylvania dataset is considerably smaller than XC, and evaluation noise would be unacceptably large if we were to subsample it.

---

> ### Author Response · Authors · 2023-11-15
> **Response to Reviewer SRto**
>
> > Given that this is a benchmark paper, and that not everybody is super familiar with the domain (bioacoustics), I would expect a more direct and clear explanation of the task being solved, the types of input given to the network, etc.
>
> Thank you for this feedback, we agree that we want to provide a helpful "table setting" for folks who aren't familiar with bioacoustics and provided overview in the Introduction and more in-depth explanation into this setting in the Appendix A.1 Why Avian Bioacoustics and A.2 Related Work. While we do describe in the main paper the base model training and inputs, along with the task description and breakdown thoroughly in the main paper, we can imagine that there would be value in having a high-level summary of all the components in either a figure, table, or brief highlight. Do you think that would help assure the reader has a clear sense of the elements of this framework? We would like to update the main paper to provide a bit more clarity for the reader.

---

> > ### Author Response · Authors · 2023-11-20
> > **Following up**
> >
> > Dear Reviewer SRto: we believe we addressed your concerns regarding the soundness of the evaluation protocol which you identified as your main concerns. Do you have any follow-up questions or concerns? If not, would you be willing to reconsider your score?
> >
> > Regarding your comment `It would be nice to report results where none of the class embeddings from training time is used`, for the setting of the Artificially Rare task, we did compare using learned weights from the output layer of the pretrained model against using class embeddings as the 'class representatives'. If that is what you had been referring to, then we would like to direct your attention to these results, shown in Figure 2 and the discussion in the 'Artificially Rare Species' paragraph in Section 4.3.
> >
> > In summary, we compared using exemplars vs. the learned representation–from our results, each model we evaluated benefits from taking advantage of the learned representation when the species was available during embedding model training, providing improved performance over using the exemplars as the query. However, depending on the application, it’s not always possible to rely on a learned representation being available (such as for regions or species not reflected in the embedding model training data).

---

> ### Comment · Reviewer_SRto · 2023-11-23
> **Review reply for the response**
>
> Thanks for the response. However, my confusion still remains. For this reason I'd like to keep my initial score.
>
> * As I mentioned in my review,
> > The evaluation benchmark retrieves relevant samples from a downstream dataset given a query, and it allows reusing class embeddings for queries learned during training. I'm not sure to get this point.
> I see no clarification regarding the use of class embeddings for "known/seen" classes. How can the identity of a sample (i.e., its class) already be known at test time? It can be predicted of course, but not mentioned in text, if I'm not mistaken.
>
> * I strongly disagree with this argument:
> > Overall, the consideration of semantic relation is irrelevant/independent when measuring label shift
>
> Semantic relation between classes/labels well impact transferability across them.

---

> > ### Author Response · Authors · 2023-11-23
> > **Clarification**
> >
> > Thank you for your feedback.
> >
> > You are correct that we evaluate the learned representation's ability to retrieve relevant audio samples given a query. We simulate a setting in which a practitioner wants to retrieve all audio clips in which a species of interest is heard in a downstream dataset of audio clips. We repeat the experiment for multiple species, collecting the ROC AUC metric each time, and then aggregating the ROC AUCs across species.
> >
> > Let us assume that the practitioner wants to retrieve audio clips for a particular bird species. They need something to compare the downstream dataset's audio clips against to rank them and hopefully surface the most relevant at the top. Usually this takes the form of one or a few example audio clips in which the bird species is heard. For BIRB we have a portion of Xeno-Canto that is held out for that purpose: it contains, for each species we evaluate on, **annotated** audio clips in which that species is heard. Those are the annotated examples we draw from to form the search query for the species of interest, and they are disjoint from both the training data *and* the downstream datasets.
> >
> > If the particular bird species the practitioner is searching for in the downstream dataset was part of the training data, they have another option: the audio representation was trained to solve a multilabel binary classification problem and it has learned one output weight vector per species in the upstream dataset. Rather than drawing examples from the held-out portion of Xeno-Canto to form a search query, we can use the learned weight vector as our basis for comparison to rank downstream audio clips.
> >
> > In summary: the labels for the downstream audio clips are not known, they are predicted (as you rightfully pointed out); what *is* known is the labels for a *disjoint* set of audio recordings that are drawn from to form the search query. If you are familiar with few-shot learning nomenclature, this is analogous to the way in which a test episode's support set is labeled but its query set is unlabeled (although to avoid any possible confusion, we point out that the "query" in the context of retrieval is a different concept from the "query set" in the context of few-shot learning: "search query" would map to "support set", and "downstream dataset whose audio clips are to be ranked" would map to "query set").
> >
> > We hope this clarifies your question and remain available to clarify further if needed.

---

### Author Response · Authors · 2023-11-22
**Manuscript update**

We again thank all reviewers for their thoughtful feedback and for recognizing some of our submission's strengths:

- Novel (N4MM: "BIRB benchmark [...] is a novel and comprehensive benchmark").
- Relevant to practitioners (SRto: "nice real-world application", "could be interesting for practitioners"; N4MM: "capturing real-world complexities", "addressing generalization challenges specific to this domain")
- Relevant to ML researchers (N4MM: "opens up opportunities for researchers to explore and innovate in the field of bioacoustics"; sG4k: "research topic is attractive to me", "real practical scenario for current machine learning field is reasonable and necessary"; sjxb: "covering various aspects of challenging ML problems")
- Comprehensive (SRto: "decent number of baselines"; NrMM: "comprehensive benchmark"; sG4k: "collected datasets are sufficient"; Sjxb: "various models are tested and benchmarked on the datasets")
- Accessible (SRto: "evaluation protocol is simple to follow/apply"; N4MM: "this work's strength lies in its use of publicly available, high-quality bioacoustic datasets")
- Informative (sjxb: "The authors find that increasing the size of networks does not translate to improved generalization, which is an interesting phenomenon")

We have thoroughly addressed each of the weaknesses/concerns you have raised, and posted an update to the manuscript which we believe addresses the main lingering reviewer concerns, which are centered on presentation:

- We included a statement in the main text (first paragraph of Section 4.2) to highlight our choice of ROC AUC as the metric for the benchmark and its relevance to the task / bioacoustics.
- We incorporated a discussion in the conclusion (Section 6) on why we think scaling up the network size does not help improve benchmark performance.
- We clarified in the main text what work was required to unify all datasets into a single, cohesive benchmark dataset (last paragraph of Section 3.1)
- We expanded the BIRB acronym the first time it is used in the abstract and in the introduction and liked to an article explaining the context behind the acronym (first page of the manuscript).

We are eagerly awaiting reviewer feedback on the proposed figure change and encourage reviewers to voice any last concerns so that we may address them.

We are confident that the strengths identified by reviewers outweigh their presentation concerns, and that any such concern can be addressed in time for the camera-ready deadline.

Thank you for helping us improve the presentation and accessibility of this important work.

The Authors

---

> ### Comment · Reviewer_SRto · 2023-11-23
> **Note on the updated paper**
>
> Thanks for updating the paper. It would have been nice to color all changes / new material in the updated version so that they can easily be tracked.

---

> > ### Author Response · Authors · 2023-11-23
> > **Changes now color-coded**
> >
> > Of course! We just uploaded an updated manuscript in which the new changes appear in teal. Note that you can also compare revisions by clicking on the "Revisions" link below the submission's author list, clicking on the "Compare Revisions" button in the top right, checking the "22 Nov 2023, 22:18 Eastern Standard Time" and "28 Sept 2023, 19:29 Eastern Daylight Time" revisions, and clicking on "View Differences". There will be a PDF comparator embedded at the bottom of the page.
> >
> > Please don't hesitate to let us know if you have any further questions or comments.

---

### Meta-Review · Area_Chair_amiG · 2023-12-07

**Metareview:**

This paper presents a benchmark for acoustic models to test their generalization and performance in distribution shift scenarios.
The authors use a dataset of ecological sounds (bird acoustics) collected by a large citizen science corpus.
The authors evaluate a number of representations/models on this benchmark and show that (a) current acoustic models do not generalize well to the benchmark; (b) scale does not solve this problem. The authors also show a baseline approach for this task.

The AC and SAC deliberated over this work. They share the reviewers' concerns around the adoption of this benchmark.

Many of the methods, such as Conformer/AudioMAE are trained on quite different domains such as LibriSpeech, Audioset. Thus, measuring their generalization to bioacoustics may not offer a lot of signal to improve these models because of the domain mismatch.
I agree with the authors that the onus of inventing novel methods is not on them. But, the current submission doesn’t answer three important questions: (1) controlling for training data, what factors matter to achieve a good performance on BIRB? Such analysis is important to provide signal for future work ; (2) how much do current methods improve on BIRB if they are trained using similar data? This helps the community infer if the poor performance on this benchmark is an artifact of the training data used by current methods. If that is indeed the case, then this benchmark’s utility is strongly tied to current training datasets. (3) Does performance on BIRB correlate with other acoustic benchmark task (either in-distribution or OOD) performance and why?

I believe addressing these concerns will strongly improve the submission quality.

**Justification For Why Not Higher Score:**

The proposed benchmark is a niche task of bird acoustics. While this is useful to measure generalization, it may not be of interest to a wide audience.

**Justification For Why Not Lower Score:**

The paper is technically correct, and interesting. The proposed benchmark is useful toe measure generalization of acoustic models.

---

### Decision · Program_Chairs · 2024-01-16

Reject